# Shaping lightwaves in time and frequency for optical fiber communication

Junho Cho [1✉], Xi Chen[1], Greg Raybon[1], Di Che[1], Ellsworth Burrows[1], Samuel Olsson [2] & Robert Tkach[1]

In optical communications, sphere shaping is used to limit the energy of lightwaves to within a certain value over a period. This minimizes the energy required to contain information, allowing the rate of information transmission to approach the theoretical limit if the transmission medium is linear. However, when shaped lightwaves are transmitted through optical fiber, Kerr nonlinearity manifests itself as nonlinear interference in a peculiar way, potentially lowering communications capacity. In this article, we show that the impact of sphere shaping on Kerr nonlinearity varies with chromatic dispersion, shaping block length and symbol rate, and that this impact can be predicted using a novel statistical measure of light energy. As a practical consequence, by optimally controlling the parameters of sphere-shaped lightwaves, it is experimentally demonstrated that the information rate can be increased by up to 25% in low-dispersion channels on a 2824 km dispersion-managed wavelength-division multiplexed optical fiber link.

[1] Nokia Bell Labs, 600 Mountain Ave, Murray Hill, NJ 07974, USA. [2] Nokia, 600 Mountain Ave, Murray Hill, NJ 07974, USA.
✉email: junho.cho@nokia-bell-labs.com

  1

The bandwidth of lightwaves was once considered an almost limitless resource for optical communications. However, due to the ever-increasing demand for higher data rates, the efficient use of bandwidth has become an important goal for today's optical communications[1]. This has led to the use of coherent transmission technology and high-order quadrature amplitude modulation (QAM), which allow more information to be contained in a single communication symbol. Lately, an information theoretic approach has been adopted for optical QAM systems to improve bandwidth efficiency by applying forward error correction (FEC) coding and probabilistic constellation shaping (PCS)[2–4] technology. This has created an intriguing phenomenon in nonlinear optical fiber communications as presented in this article, which was never seen in other communications systems.

An implementation of PCS called sphere shaping[5–17], notably important in both theory and real-world applications, only creates the blocks of communication symbols with the total energy not exceeding a certain limit. This fundamentally minimizes the average energy of a spectro-temporal block of lightwaves to contain a given amount of information. When the lightwaves are transmitted through a linear medium, increasing the shaping block length towards infinity allows the information rate to approach the theoretical limit[18]. However, when the lightwaves are transmitted over optical fiber, the sphere shaping can increase Kerr nonlinearity[19–22] and lower the nonlinear capacity[21] while retaining its fundamental energy efficiency[5–10]. The unique way that the sphere shaping constrains energy changes the statistical properties and temporal structure of the shaped lightwaves differently depending on the shaping block length, and accordingly the manifestation of Kerr nonlinearity varies[8,11,12,22]. For this reason, there have been several recent approaches to optimizing the shaping block length to mitigate Kerr nonlinearity[8,11,12,15,16,23].

In addition to the shaping block length, the symbol rate at which communication symbols can change values also affects Kerr nonlinearity. This is because the symbol rate determines the way a continuum of the lightwaves, in time and frequency, is divided into small blocks of symbols in today's densely packed wavelength-division multiplexing (WDM) systems. Namely, the frequency bandwidth of symbols increases in proportion to the symbol rate while their time duration decreases inversely. Therefore, changing the symbol rate changes both the spectral and temporal properties of the lightwaves. For the traditional unshaped lightwaves that carry statistically independent and identically distributed (i.i.d.) symbols, the influence of the symbol rate has been well established through simulation, experiment, and analysis[24,25]. However, in the case of sphere-shaped lightwaves, the presence of a temporal structure of energy invalidates the i.i.d. assumption and makes it difficult to study the Kerr nonlinearity analytically. Analytical approaches to take into account the structure of lightwaves have so far been successful up to one symbol[26,27], but extending the analysis to structures spanning many symbols seems mathematically daunting. To quantify the effect of large temporal structures of lightwave on Kerr nonlinearity, empirical approaches are being taken in rapidly growing recent studies[8,15–17,23]. However, there has been no study on whether or how the symbol rate affects this quantification.

In this article, we show that for a given fiber link, shaped symbols with the same block length can be affected differently by Kerr nonlinearity depending on the symbol rate. Furthermore, using a novel statistical measure of light energy, we provide a comprehensive picture showing the relationship between the spectro-temporal block size of sphere-shaped lightwaves, chromatic dispersion of fiber, and Kerr nonlinearity. This comprehensive picture elucidates the seemingly inconsistent results observed between experiments performed independently with different settings, without which shaped lightwaves may appear to exhibit peculiar behavior. Not only does our finding contribute to an intrinsic understanding of the characteristics of sphere-shaped lightwaves, it also allows the most efficient use of optical bandwidth by optimizing the parameters of sphere shaping in WDM systems. By adjusting the spectro-temporal block size of sphere shaping, we experimentally demonstrate that the effective signal-to-noise ratio (SNR) increases by up to 1.1 dB and the net data rate (NDR) increases by up to 25% in low-performing channels in a 3.7-THz-wide full C-band transmission system.

## Results

**Sphere shaping of lightwaves**. We consider a polarization-division multiplexed (PDM) $M^2$-ary QAM system. A digital shaping encoder at the transmitter produces blocks of amplitudes chosen from an equally spaced numerical alphabet $\mathscr{A} = \{1, 3, \ldots, 2M - 1\}$ in arbitrary units. Each amplitude is multiplied by an equiprobable sign in $\{+1, -1\}$, resulting in a probability distribution over $\{\pm 1, \pm 3, \ldots, \pm(2M - 1)\}$ symmetric around the origin. The symmetry of the probability allows for legitimate analysis with only positive amplitudes, and hence we omit the sign throughout this article for descriptive purposes (but in simulations and experiments, equally distributed positive and negative signs are used). In our system, four *consecutive* amplitudes constitute one dual-polarization symbol, as has been done, e.g., in[16,17], that is transmitted with a symbol period of $T_{Sym}$ and a symbol rate of $R_{Sym} = 1/T_{Sym}$. We therefore set up the encoder to produce a $4n$-long amplitude block over $n$ symbol periods. Let us denote this amplitude block as $\mathbf{a} = [a_1, \ldots, a_{4n}]$ with $a_i \in \mathscr{A}$ for $i = 1, \ldots, 4n$ and the corresponding symbol block as $\mathbf{x} = [x_1, \ldots, x_n]$ with dual-polarization symbols $x_i = \left[a_{4i-3}, a_{4i-2}, a_{4i-1}, a_{4i}\right] \in \mathscr{A}^4$ for $i = 1, \ldots, n$. Then, in traditional QAM systems, the amplitude block $\mathbf{a}$ (and hence the symbol block $\mathbf{x}$) can have a maximum total energy of $E^*_{\text{Unshaped}} = \max \|\mathbf{a}\|^2 = 4n(2M - 1)^2$ when all amplitudes are at their maximum, where $\|\cdot\|$ denotes Euclidean norm. On the other hand, sphere shaping imposes a limit $E^*_{\text{Shaped}}$ on the total energy such that only amplitudes that jointly fulfill $\|\mathbf{a}\|^2 \leq E^*_{\text{Shaped}}$ are created by the encoder, where $E^*_{\text{Shaped}} < E^*_{\text{Unshaped}}$ in general. This method is called sphere shaping[5–17] because if every shaped block is plotted as a point in $4n$-dimensional signal space, with the $i$-th amplitude being the position of the point on the $i$-th coordinate axis, the points are distributed uniformly over a set of $4n$-dimensional square lattice points that lie on or contained in a $4n$-dimensional (hyper-) sphere of radius $\sqrt{E^*_{\text{Shaped}}}$ (due to the symmetry by equiprobable signs). As we decrease the limit $E^*_{\text{shaped}}$, more combinations of amplitudes are not allowed to be created, hence less information can be contained per block. This is because the maximum number of information bits per block is given by $\lfloor \log_2 N \rfloor$ if there are $N$ possible combinations of amplitudes, where $\lfloor \cdot \rfloor$ denotes the floor function. The maximum information rate in bits per dual-polarization symbol is then given by $R = \lfloor \log_2 N \rfloor / n + 4$, where the addition by 4 accounts for four sign bits per symbol. The average energy of $\mathbf{a}$ to achieve $R$ with sphere shaping decreases with increasing block length $4n$ (see, e.g.,[8,16]), achieving a theoretical minimum average energy as $n \to \infty$. We refer to the reduction in average energy of $\mathbf{a}$ by shaping as the fundamental shaping efficiency in this article. Figure 1(a) shows the probability distributions of the total energy in a block of length $n = 5$. Compared to the unshaped PDM 16-QAM (left figure) with $R = 8$ and $E^*_{\text{Unshaped}} = 180$, sphere shaping (right figure) substantially reduces the maximum energy

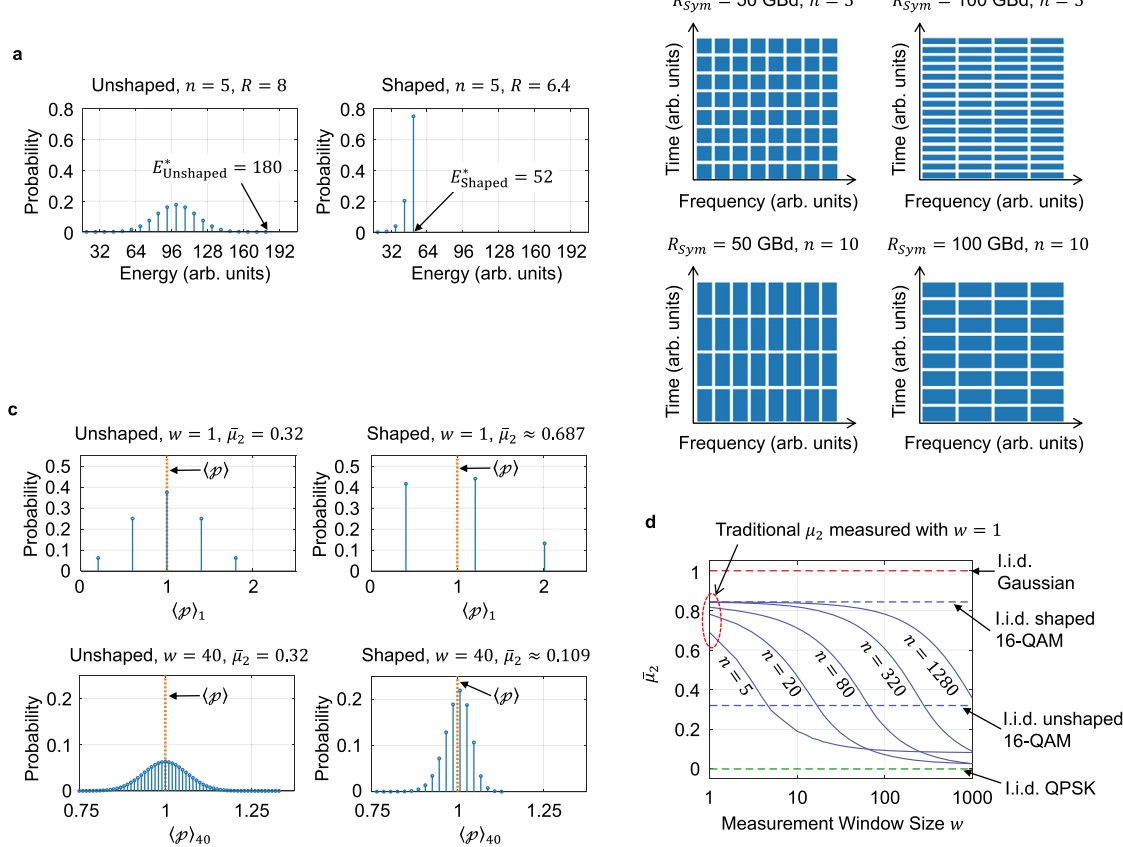

**Fig. 1 Comparison of the power deviation between unshaped and shaped lightwaves. a** Probability distribution of the total energy in a block of the unshaped (left figure) and shaped (right figure) symbols. The maximum energy is significantly lower in the shaped symbols than in the unshaped symbols. **b** Division of the continuum of lightwaves in time and frequency. The probability distributions shown in (**a**) appear in each of the rectangular blocks that divide the lightwaves. **c** Probability distribution of the normalized power $\langle \wp \rangle_w$ of the unshaped (left figures) and shaped (right figures) symbols, measured with a sliding window of length $w = 1$ (upper figures) and 40 (lower figures). **d** The windowed second central moment $\bar{\mu}_2$ as a function of the measurement window size $w$. As $w$ increases, the moment decreases only for the shaped symbols. All shaped symbols in Fig. 1 achieve $R = 6.4$.

to $E_{\text{Unshaped}}^* = 52$ (71% reduction) at the expense of only 1.6 bit reduction in $R$ (20% reduction). In a densely packed WDM system with identical channel configurations, such probabilistic energy distributions as in Fig. 1(a) are observed within each rectangular block that divides lightwaves in the time-frequency plane as shown in Fig. 1(b), where the width and height of the block are determined by both $R_{Sym}$ and $n$. While the shaping block length $n$ or the distribution of energy (cf. Figure 1(a)) has been optimized in existing studies[8,11,12,15–17,23] to mitigate non-linear interference (NLI), the spectro-temporal region where the distribution is found (cf. Figure 1(b)) has never been noted previously. In the following sections, we will see that it is the distribution of light energy in all aspects of probability, time, and frequency that determines the manifestation of Kerr nonlinearity as NLI, and thus $R_{Sym}$ and $n$ must be controlled simultaneously to minimize NLI.

**Power deviations in sphere-shaped lightwaves.** In long-haul optical fiber links, the NLI generated from i.i.d. symbols behaves similarly to additive white Gaussian noise (AWGN)[28–30]. The average NLI power increases in proportion to the cube of the average symbol power as[28–30]

$$\langle P_{NLI} \rangle = \eta \langle \|x\|^2 \rangle^3, \qquad (1)$$

where $\langle \cdot \rangle$ denotes statistical averaging, $\eta$ is the NLI coefficient determined mostly by link parameters and to some extent by

modulation. Denoting the symbol power normalized to have unit mean by $\wp \triangleq \|x\|^2 / \langle \|x\|^2 \rangle$, $\eta$ increases as the *central moment* $\mu_n$ of $\wp$ increases, where $\mu_n$ is defined as

$$\mu_n = \langle (\wp - 1)^n \rangle. \qquad (2)$$

Namely, $\mu_n$ quantifies the $n$-th order deviation of instantaneous $\wp$ from average $\langle \wp \rangle = 1$, and the more $\wp$ deviates from $\langle \wp \rangle$, the greater the NLI. Note that the instantaneous and average powers are implicitly measured over a symbol period $T_{Sym}$ and infinite time, respectively.

A question arises here as to whether measuring the instantaneous power over a single symbol period is appropriate to study the light propagation effect, since the shaped symbols have a unique temporal energy structure that, while propagating through optical fiber, results in a different evolution compared to i.i.d. symbols. The presence of the temporal energy structure in dispersive medium suggests that a longer time period than $T_{Sym}$ may better characterize the light propagation effect[31,32]. To find an appropriate time to measure the instantaneous power, we define a new statistical figure of merit called the windowed central moment of $\wp$ as

$$\bar{\mu}_n = \langle (\langle \wp \rangle_w - 1)^n \rangle \cdot \underbrace{(2w)^{n-1}}_{(a)}, \qquad (3)$$

where $\langle \cdot \rangle_w$ denotes a moving average filter with a sliding window of length $w$ symbols (with a sliding step size of one symbol). Namely, $\bar{\mu}_n$ quantifies the $n$-th order power deviation using the instantaneous

power measured over $w$ symbol periods, with $w$ being a free parameter that will be optimized in the following section. Since $\langle \rlap{/}p \rangle_w$ approaches 1 as $w$ increases, the term (a) on the right-hand side ensures that $\bar{\mu}_n$ remains the same regardless of $w$, if $\rlap{/}p$ is i.i.d. Here, the factor 2 compensates for $\langle \rlap{/}p \rangle_w$ being averaged over 2 polarizations. Without the term (a), the second windowed central moment $\bar{\mu}_2$ equals the energy dispersion index that was lately suggested[32].

Figure 1(c) shows the probability distributions of $\langle \rlap{/}p \rangle_w$ for PDM 16-QAM symbols without (left figures) and with (right figures) sphere shaping ($n = 5$). When the instantaneous power is measured with $w = 1$ as conventionally done (upper figures), sphere shaping appears to make the instantaneous power more spread out around the average power (note the increase of $\mu_2$ from 0.32 to 0.687 after sphere shaping). If we interpret this using the existing analytical model developed for i.i.d. symbols[30,33], it implies that sphere shaping increases NLI compared to the unshaped lightwaves[17,34]. On the contrary, if $n$ is not large as shown in Fig. 1(c), recent simulations and experiments[8,15,16,23] show that sphere shaping does not necessarily increase and can even decrease NLI compared to the unshaped lightwaves. The contradiction between the model and empirical observations may be attributed to the fact that, when energy structures exist in time, the power deviation measured with $w = 1$ does not represent a proper statistical measure to analyze lightwaves. This is a plausible explanation given that, as shown in Fig. 1(c), when $w$ increases from 1 (upper figures) to 40 (lower figures), $\langle \rlap{/}p \rangle_w$ becomes concentrated near $\langle \rlap{/}p \rangle$ for the shaped symbols (right figures) due to the block-wise energy constraint, whereas it maintains the variance for the unshaped symbols (left figures). Depicted in Fig. 1(d) is $\bar{\mu}_2$ as a function of $w$. As $w$ increases, $\bar{\mu}_2$ remains constant for i.i.d. symbols (dashed lines), but it decreases for the finite-length shaped symbols (solid lines). Therefore, if it is the power deviation measured over a longer period than $T_{Sym}$ that determines the manifestation of Kerr nonlinearity, it may be possible to make the analytical model and empirical observations consistent by replacing the conventional statistical measure in Eq. (2) with the new measure in Eq. (3), as we will investigate further in the following sections.

**Optimization of sphere shaping parameters in the time-frequency plane**. We first study the influence of sphere shaping on NLI using split-step simulations[35,36]. We transmit symbols in a total bandwidth of 100 GHz centered at 193.4 THz (1550.1 nm) over $N_{Ch}$ channels that evenly divide the total bandwidth, with $N_{Ch} = 1, 2, 4, 8, 16, 32, 64$. The symbols are transmitted in each channel at a rate of $R_{Sym} = 88/N_{Ch}$ GBd, ranging from 1.375 to 88 GBd, leaving 12% spectral margins for root-raised cosine (RRC) pulse shaping with a roll-off factor of 0.1. Sphere shaping is performed in each channel with $n = 5, 10, 20, 40, 80, 320, 1280$, and 5120, with a fixed $R = 6.4$ bits per dual-polarization symbol using 16-QAM. For comparison, i.i.d. shaping is also performed on 16-QAM using a Maxwell-Boltzmann distribution with 6.4 bits of entropy per dual-polarization symbol. Therefore, all the WDM configurations under test send data at the same rate in the same total bandwidth. We assume, throughout the article, the use of a rate-0.8 field-programmable gate array (FPGA)-verified spatially-coupled low-density parity-check (LDPC) code[37] for forward error correction (FEC), which has a normalized generalized mutual information (NGMI) threshold[38,39] of $NGMI^* = 0.86$ for error-free decoding. With this, the same total information rate of 422.4 Gb/s can be achieved by all the WDM configurations. Four links with regular 60 km spans are constructed by standard single-mode fiber (SSMF) of length $L_{SSMF}$ km followed by dispersion compensating fiber (DCF) of length $L_{DCF} = 60 -$

**Table 1 Link configurations in which the impact of sphere shaping is evaluated.**

| Link Name | $L_{SSMF}$ | $L_{DCF}$ | Net dispersion $D_{Span}$ per span |
|---|---|---|---|
| Link A | 49.36 km | 10.64 km | 0 ps/nm |
| Link B | 49.98 km | 10.02 km | 60 ps/nm |
| Link C | 51.83 km | 8.17 km | 240 ps/nm |
| Link D | 60 km | 0 km | 1033.8 ps/nm |

$L_{SSMF}$ km in each span (with dispersion coefficients of 17.24 and $-80$ ps/nm/km, respectively), as shown in Table 1. An Erbium doped fiber amplifier (EDFA) recovers the launch power after every span with 4.5 dB noise figure. Complete details of the simulation setup are given in the Methods section.

Figure 2(a) shows the effective SNR as a function of the symbol rate (on x-axis) and shaping block duration (on y-axis), defined as

$$SNR_{Eff} = \frac{\langle \|x\|^2 \rangle}{\langle P_{ASE} \rangle + \langle P_{NLI} \rangle}, \tag{4}$$

where $\langle P_{ASE} \rangle$ is the average amplified spontaneous emission (ASE) noise power. The signal, ASE, and NLI powers are all measured over the total bandwidth of 100 GHz. The contour lines are obtained by interpolating 63 simulation points (red dots). The top points at each $R_{Sym}$ represent i.i.d. shaping, so their y-axis values are not exact values but merely represent very large numbers. The y-axis values for all other points are exact. The bottom points at each $R_{Sym}$ represent $n = 5$, but for the same $n$, the shaping block duration in nanoseconds on y-axis varies with $R_{Sym}$. The launch power is optimized for each point to maximize $SNR_{Eff}$. The effective SNR is evaluated at the transmission distances where the NGMI is near the threshold. Looking at Fig. 2(a), we notice the following: (i) like the i.i.d. uniform 16-QAM[25], the optimal $R_{Sym}$ for i.i.d. shaping of 16-QAM to maximize $SNR_{Eff}$ (yellow stars) decreases as the total net dispersion $D_{Total}$ increases, (ii) reducing $n$ improves $SNR_{Eff}$ if $R_{Sym}$ is near optimal (red stars), and (iii) the optimal $R_{Sym}$ increases as $n$ decreases (compare, e.g., the yellow and red stars). The highest $SNR_{Eff}$ at the smallest $n$, however, does not necessarily maximize the end-to-end communication performance measured in NGMI, as shown in Fig. 2(b), due to the complex interplay with the fundamental shaping efficiency that decreases as $n$ decreases. In general, (i) the optimal $n$ for NGMI tends to increase with $D_{Total}$, and (ii) the optimal $R_{Sym}$ for NGMI is higher than the optimal $R_{Sym}$ for $SNR_{Eff}$.

We scrutinize the nonlinear effect by extracting the NLI coefficient $\eta$ from the simulation results as

$$\eta = \left[ \frac{\langle \|x\| \rangle^2}{SNR_{Eff}} - \langle P_{ASE} \rangle \right] / \langle \|x\|^2 \rangle^3, \tag{5}$$

derived from Eqs. (1) and (4). The result is shown in Fig. 2(c-1). We factorize $\eta$ as

$$\eta = \eta^\infty \times \Delta\eta, \tag{6}$$

where $\eta^\infty$ represents the NLI coefficient of i.i.d. shaping, cf. Figure 2(c-2), and $\Delta\eta \leq 1$ is a correction factor for finite-length shaping, cf. Figure 2(c-3). Furthermore, from another set of simulations using the same WDM settings as above, yet with only one channel being transmitted in each setting, we evaluate $\eta_{SPM}$ produced solely by self-phase modulation (SPM)[40,41], cf. Figure 2(c-4). Then, by subtracting the SPM contribution from the full-fledged simulation results, we obtain $\eta_{XPM}$ produced solely by cross-phase modulation (XPM)[42,43], cf. Figure 2(c-7).

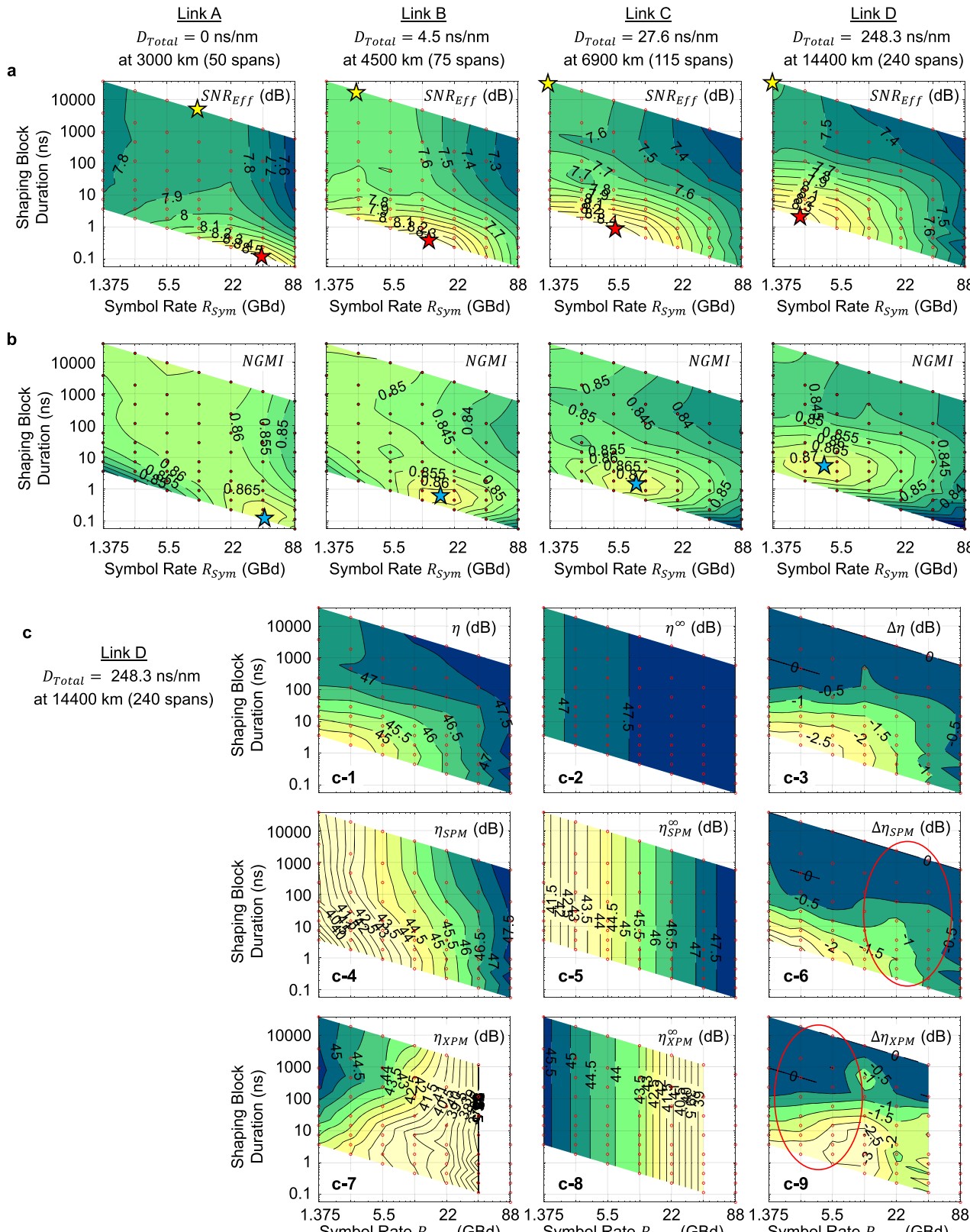

**Fig. 2 Split-step simulation with various spectro-temporal division of lightwaves. a** Effective SNR drawn as a function of the symbol rate (spectral division) and shaping block duration (temporal division). **b** NGMI showing the end-to-end performance, produced by the complex interaction of the propagation effect and the fundamental energy efficiency of the underlying sphere shaping. **c** NLI coefficient $\eta$ and its decompositions into SPM contribution $\eta_{SPM}$ and XPM contribution $\eta_{XPM}$, and into factors due to i.i.d. shaping $\eta^\infty$ and finite-length shaping $\Delta\eta$. Source data for Fig. 2 are provided with this paper.

Looking at Fig. 2(c-5), (c-8), as $R_{Sym}$ increases, $\eta_{SPM}^\infty$ increases whereas $\eta_{XPM}^\infty$ decreases, as a consequence of the relative increase in channel bandwidth within a fixed total bandwidth (see also[25]), and thus the combined effect $\eta^\infty$ changes just marginally, cf. Figure 2(c-2). The influence of $R_{Sym}$ on $\eta^\infty$ is expected to decrease further as the modulation order increases and the shaped signal approaches continuous Gaussian. Looking at the finite-length effect in Fig. 2(c-6), (c-9), the influence of $\Delta\eta_{SPM}$ and $\Delta\eta_{XPM}$ on $SNR_{Eff}$ is prominent only near the red circled areas, where their base coefficients $\eta_{SPM}^\infty$ and $\eta_{XPM}^\infty$ are large. Areas far from the circled areas have a negligible impact on $SNR_{Eff}$. Combining $\Delta\eta_{SPM}$ and $\Delta\eta_{XPM}$, $\Delta\eta$ changes much more than $\eta^\infty$, indicating that $n$ has a much greater impact on $SNR_{Eff}$ than $R_{Sym}$ near the optimal point that maximizes $SNR_{Eff}$. This leads to $SNR_{Eff}$ of Fig. 2(a). More results are provided in Supplementary Figs. 1–4.

The window size for $\bar{\mu}_2$ is then determined in a similar way to[32] by calculating the Pearson correlation coefficient $\rho(\eta, \bar{\mu}_2)$ that quantifies how much $\eta$ is correlated with $\bar{\mu}_2$. This is done separately for SPM and XPM, as shown in Fig. 3(a) at 240 spans in Link D. The red lines show the linear fits of the optimal window sizes $w_{SPM}^*$ and $w_{XPM}^*$ that produce the greatest $\rho(\eta_{SPM}, \bar{\mu}_2)$ and $\rho(\eta_{XPM}, \bar{\mu}_2)$, respectively, which are given by

$$w_{SPM}^* \approx 2R_{Sym}B_{Ch}|\beta_2|L_{Span}N_{Span} \qquad (7)$$

and

$$w_{XPM}^* \approx 2R_{Sym}B_{Ch}\sqrt{N_{Ch}/0.88}\,|\beta_2|L_{Span}N_{Span} \qquad (8)$$

in number of symbols. Here, $R_{Sym}$ is in $\mathrm{s}^{-1}$, $B_{Ch} \approx R_{Sym}$ is the channel bandwidth in $\mathrm{s}^{-1}$, $\beta_2$ is the dispersion coefficient in $\mathrm{s}^2/\mathrm{m}$, $L_{Span}$ is the span length in m, and $N_{Span}$ is the number of spans. In Eq. (8), the division by 0.88 is due to channel spacing, and the factor $\sqrt{N_{Ch}}$ indicates that $w_{XPM}^*$ increases with the number of channels but only in proportion to its square root. This is consistent with the fact that more symbols from the copropagating channels are involved in nonlinearity as the number of channels increases but more distant channels contribute less to nonlinearity. Note that Eqs. (7) and (8) are valid for all the systems under test (cf. Supplementary Figs. 5–8). Plugging the average symbol power $\langle\|x\|^2\rangle$ and the windowed central moments $\bar{\mu}_2$ and $\bar{\mu}_3$ obtained with $w_{SPM}^*$ and $w_{XPM}^*$ into a state-of-the-art analytic model known as the enhanced Gaussian noise (EGN) model[30,33], we obtain $SNR_{Eff}$ as shown in Fig. 3(b) (see the Methods section for more details). The EGN model assumes i.i.d. amplitudes and phases of symbols, and hence is not accurate for lightwaves with local energy structures. There is a recently developed analytical model[26,27] that extends the EGN model to account for energy structures present over one symbol period, but extending this further to energy structures spanning tens to thousands of symbol periods that we deal with in this work seems mathematically intractable. Therefore, we allow for model mismatch by using the classical EGN model, but improve the accuracy of evaluating structured lightwaves (green solid lines in the figure) by replacing $\mu_2$ and $\mu_3$ with optimized $\bar{\mu}_2$ and $\bar{\mu}_3$ (i.e., obtained with $w_{SPM}^*$ and $w_{XPM}^*$). This provides good agreement with the split-step simulation results (black dashed lines) in a wide range of conditions (cf. Supplementary Figs. 9, 10). This shows that the energy structure, whether short or long, can be taken into account with manageable complexity when analyzing the light propagation effect.

**Demonstration of optimal sphere shaping through full C-band transmission experiment.** We experimentally demonstrated the optimal sphere shaping for maximum NDR in a full C-band transmission system shown in Fig. 4(a)[44]. The dependence of the optimal sphere shaping on dispersion is conveniently verified in a recirculation loop that accumulates varying dispersions over frequency. The loop consists of 7 spans of 40.3 km (on average) fiber, 1 span of which (span #4) is SSMF (with 0.092 ps/nm²/km dispersion slope) and the rest are anomalous-dispersion fiber (with –2.47 ps/nm/km dispersion and –0.1026 ps/nm²/km dispersion slope, at 1550.1 nm). We transmit 37 100 GHz-wide channels in the C-band over 10 loops (2824 km). Figure 4(b) shows the accumulated dispersion as a function of distance for several selected channels. We denote the channels at 192.1–195.7 THz sequentially by Ch#1 to Ch#37. Then, Ch#8 at 192.8 THz (solid red line) undergoes zero net dispersion after every loop, and the farther away from Ch#8 the channel undergoes greater absolute net dispersion, reaching up to $|D_{Total}| = 4.85$ ns/nm at Ch#37.

Three independent streams of symbols are transmitted over the loop, one of which is loaded on the channel under test (CUT), the other two on the even- and odd-indexed interfering channels that are fully decorrelated by delay fibers of distinct lengths (10, 20, …, 190 m for odd channels, and 10, 20, …, 180 m for even channels, cf. Figure 4(a)). At the transmitter (TX), all interfering channels are loaded with the same $E_{Shaped}^*$, $n$, and $R_{Sym}$ as the CUT. At the receiver (RX), commonly used coherent digital signal processing (DSP) is performed offline to recover the transmitted symbols. Complete details of the experimental setup are given in the Methods section.

We implement up to 20 combinations of $n$ and $R_{Sym}$ by using $n = 5, 10, 20, 80, 320$, and $R_{Sym} \approx 9.8, 19.7, 39.4, 78.8$ GBd. On each of the 100 GHz-wide optical channels in our experimental system, we emulate the four $R_{Sym}$ above by performing digital subcarrier multiplexing (DSM)[45,46] with $N_{SC} = 1, 2, 4, 8$ digital subcarriers on an optical carrier modulated at 78.8 GBd, as shown in Fig. 4(c). RRC pulse shaping is performed on each subcarrier with a roll-off factor flexibly adjusted between 0.05 and 0.1 depending on $N_{SC}$. With this, we ensure in a back-to-back experiment that $SNR_{Eff}$ averaged over subcarriers differs by no more than 0.2 dB between DSM of all $N_{SC}$ at the same optical SNR (OSNR), thereby circumventing the bandwidth limitations of the transmitter favoring $N_{SC} = 1$ (see the Methods section for more details).

For each pair of $n$ and $R_{Sym}$, the NDR achieved by sphere shaping of 16-QAM is determined by finding the maximum $R^* \in \{5.2, 5.6, \ldots, 7.6\}$ varied in increments of 0.4, which yields an NGMI greater than $NGMI^* = 0.86$. The NDR per optical carrier is then given by

$$NDR = 4 \times \left[R^*/4 - m(1 - R_C)\right]R_{Sym}N_{SC}R_{Pi} \qquad (9)$$

in Gb/s, where the multiplication by 4 in the right-hand side accounts for 4 amplitudes per symbol time, $m = \log_2 M$, and $R_{Pi} = 47/48$ is the pilot ratio used to assist coherent DSP. Instead of $R = 4.8$ for shaped 16-QAM, we use uniform quadrature phase-shift keying (QPSK), since they produce the same NDR. Combined with up to 20 configurations of $n$ and $R_{Sym}$, changing $R$ for each channel to maximize the NDR implies that a huge number of measurements are made across the 37 channels in C-band for experimental verification of optimal sphere shaping.

We present the result obtained with the launch power (defined as the total power for the full C-band) of 13 dBm in this article, although the results are obtained with various launch powers from 8 to 13 dBm in 1 dBm steps. The power excursion across the C-band is maintained within 4 dB after 10 loops by adjusting the inline dynamic gain equalizers (DGEs, cf. Figure 4(a)). The power excursions due to experimental constraints (e.g., a small power excursion caused by coarse attenuation granularity of the DGEs

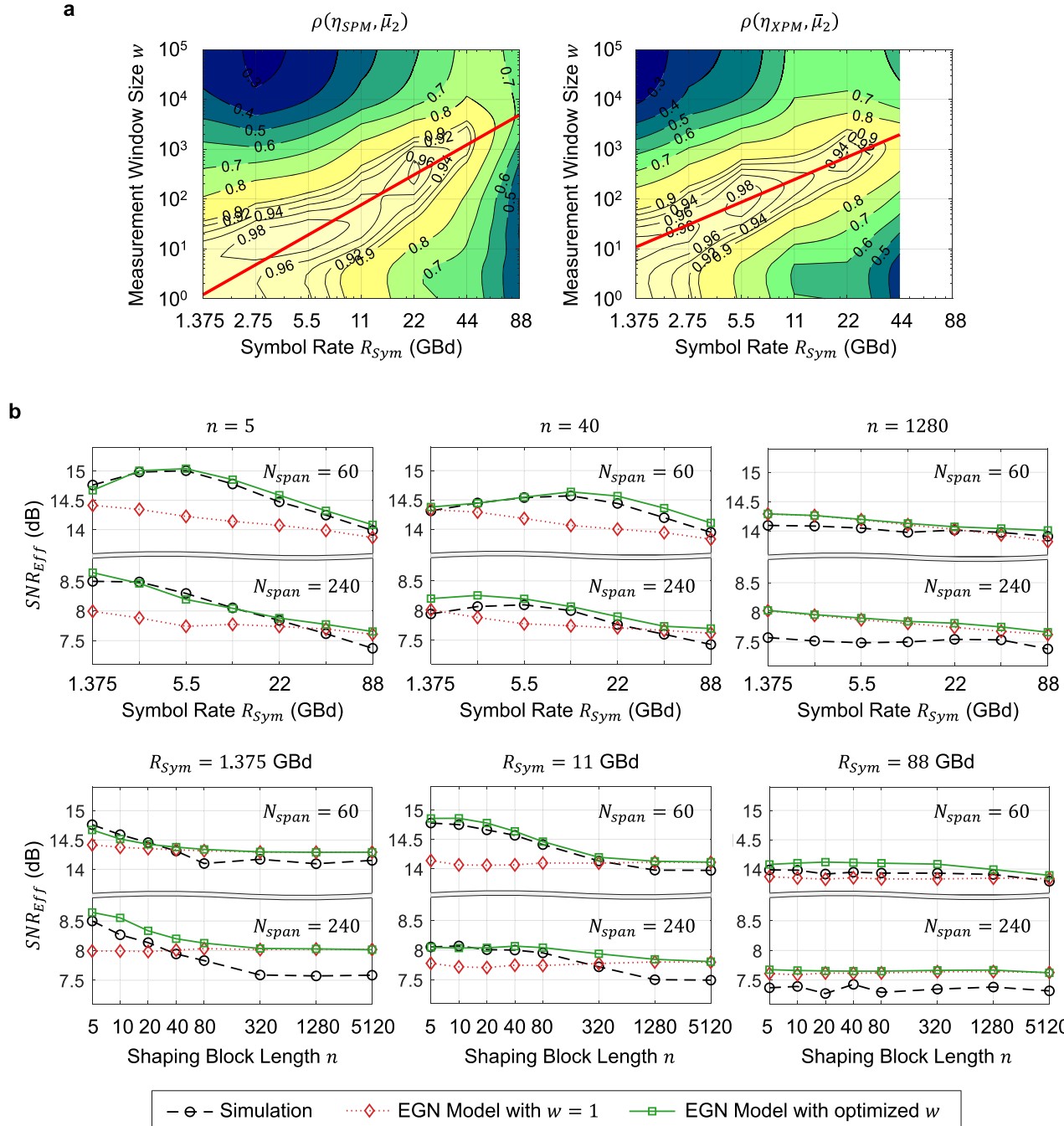

**Fig. 3 The optimal window size for instantaneous power measurement and $SNR_{Eff}$ predicted by the EGN model using it. a** Pearson correlation coefficients between $\eta_{SPM}$ and $\bar{\mu}_2$ (left figure), and between $\eta_{XPM}$ and $\bar{\mu}_2$ (right figure) with various window sizes $w$, obtained at 240 spans in Link D. The red lines are obtained using Eqs. (7) and (8), estimated to produce the greatest correlation. There is no XPM at $R_{Sym} = 88$ GBd, since only one channel is transmitted. **b** $SNR_{Eff}$ predicted by the EGN model, obtained with the optimal $w^*_{SPM}$ and $w^*_{XPM}$ (green solid lines) and traditional $w = 1$ (red dotted lines), in comparison with the split-step simulation results (black dashed lines) at 60 and 240 spans in Link D. Source data for Fig. 3(a) are provided with this paper.

results in an increasing power excursion as the number of loops increases) are considered to be the most important contributor to discrepancy in validation of theory. Typical recovered constellations are shown in Fig. 4(a) next to the RX. Figure 5(a) and (b) show, respectively, the optimal $n^*$ and $R^*_{Sym}$ that jointly maximize NDR in each channel. In general, $n^*$ tends to increase as $|D_{Total}|$ (green solid line) increases, which is consistent with the simulation result in Fig. 2(b). $R^*_{Sym}$ tends to decrease from 78.8 to 39.4 GBd when $|D_{Total}|$ increases from approximately 2 to 4 ns/

nm, and this trend of change also matches the simulation result in Fig. 2(a). Using the optimal $n^*$ and $R^*_{Sym}$ of Fig. 5(a, b), $SNR_{Eff}$ of Fig. 5(c) (orange circles) is obtained. Note that greater $SNR_{Eff}$ than those in Fig. 5(c) are obtained with $n$ smaller than $n^*$, but they provide smaller NDRs due to the reduced fundamental shaping efficiency. Also shown in Fig. 5(c) is the $SNR_{Eff}$ of the shaping with the longest $n = 320$ (blue triangles), chosen as a benchmark for its best performance in linear channels, at a symbol rate of $R_{Sym} = 39.4$ GBd that represents a widely used

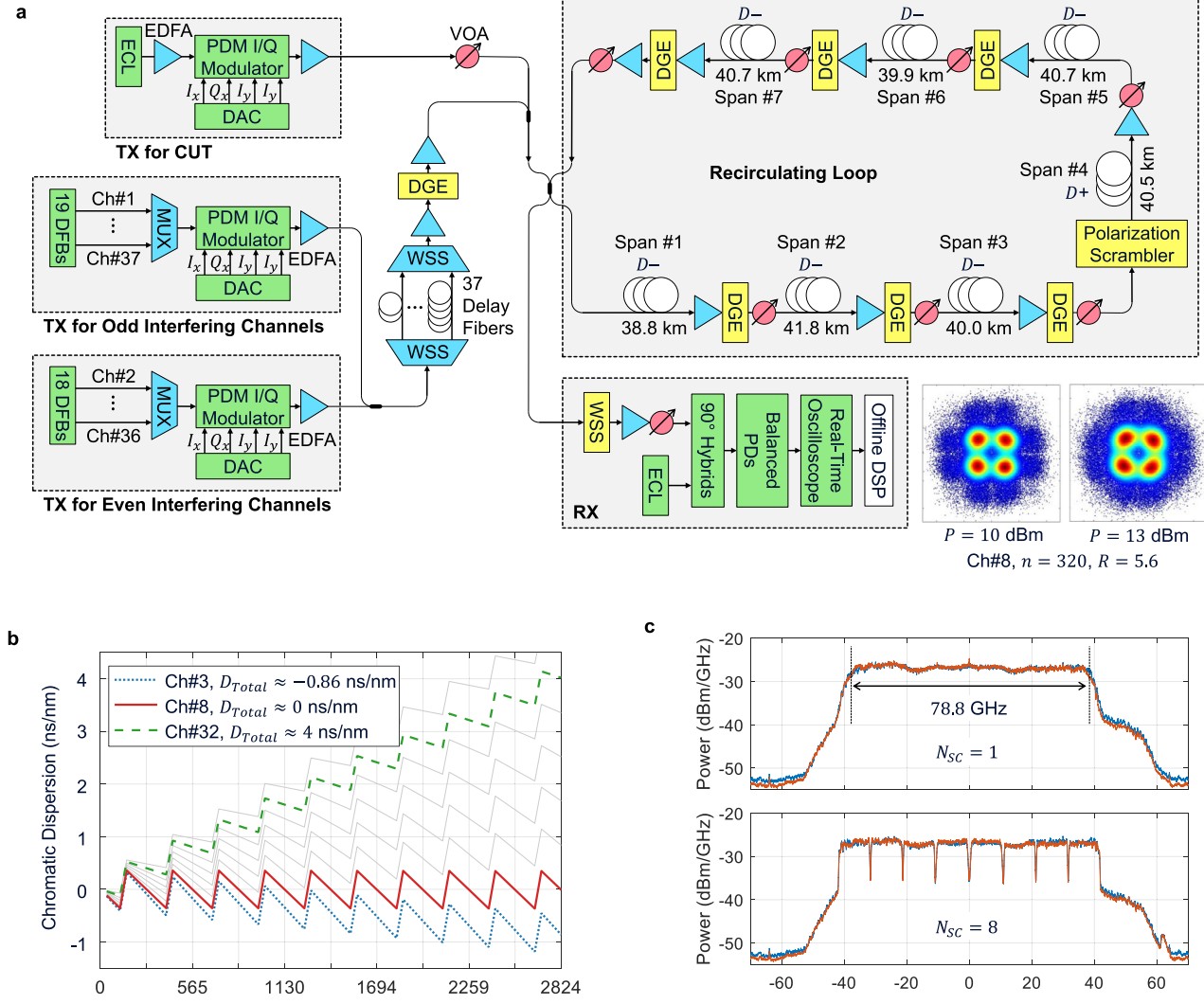

**Fig. 4 Experimental system to verify the performance of optimal sphere shaping in full C-band transmission. a** Schematic of the experimental setup. Inset: typical recovered constellations at the receiver. **b** Dispersion map of various WDM channels over the optical fiber link. **c** Typical received power spectral densities when a single (upper figure) and eight (lower figure) digital subcarriers are transmitted.

value in deployed coherent systems. Near the zero-dispersion regime at 192.8 THz, the benchmark shaping does not produce an NGMI greater than $NGMI^*$ with any $R$, hence the channels are modulated with QPSK (blue pluses in Fig. 5(c)). Figure 5(d) shows the NDRs achieved by the optimal $n^*$ and $R^*_{Sym}$ (orange circles), and by the benchmark shaping (blue triangles), where the NLI-optimized parameters offer higher NDRs than the linear-channel optimal benchmark widely in low-dispersion channels. The NDR increase by the NLI-optimized parameters reaches 25% near the zero-dispersion regime (cf. red arrow in Fig. 5(d)) and 12.1% on average over the underperforming 20 channels with low dispersion. The total NDR increase in the C-band reaches 6.4% (12.86 Tb/s as compared to 12.09 Tb/s of the benchmark, obtained as the sum of the NDRs of all 37 channels in Fig. 5(d)). While the previous works[8,11,12,16,23] optimized only $n$ to observe some gains in $SNR_{Eff}$ and NDR over specific links (e.g., for single-span links), joint optimization of $n$ and $R_{Sym}$ in this work produces significantly larger gains and allows these gains to be achieved over a much wider variety of links.

## Discussion

As seen from the comparison of the finite-length and i.i.d. shaped lightwaves under various conditions, finite-length sphere shaping reduces Kerr nonlinearity. To explain this phenomenon, we hypothesized that there is a specific time window for quantifying the power deviation of lightwaves to best describe the propagation effect and that, when measured with this time window, the power deviation is reduced by the small spherical energy structures in the shaped lightwaves. We provided three evidences that support this hypothesis. First, the many horizontal line segments of the NLI contour $\Delta\eta$ observed in wide areas of Fig. 2 and Supplementary Figs. 1–4 indicate that Kerr nonlinearity depends on the spherical energy structure present in absolute time rather than in the number of symbols. Second, Eqs. (7) and (8) show that the measurement window size for the instantaneous power agrees remarkably well with the current theoretical understanding of SPM and XPM in dispersive medium. Third, the analytical results obtained with such an optimized time window fairly match the numerical simulation results, as shown in Fig. 3 and Supplementary Figs. 9 and 10. Nevertheless, we cannot completely rule out the possibility that there may be other, yet undiscovered,

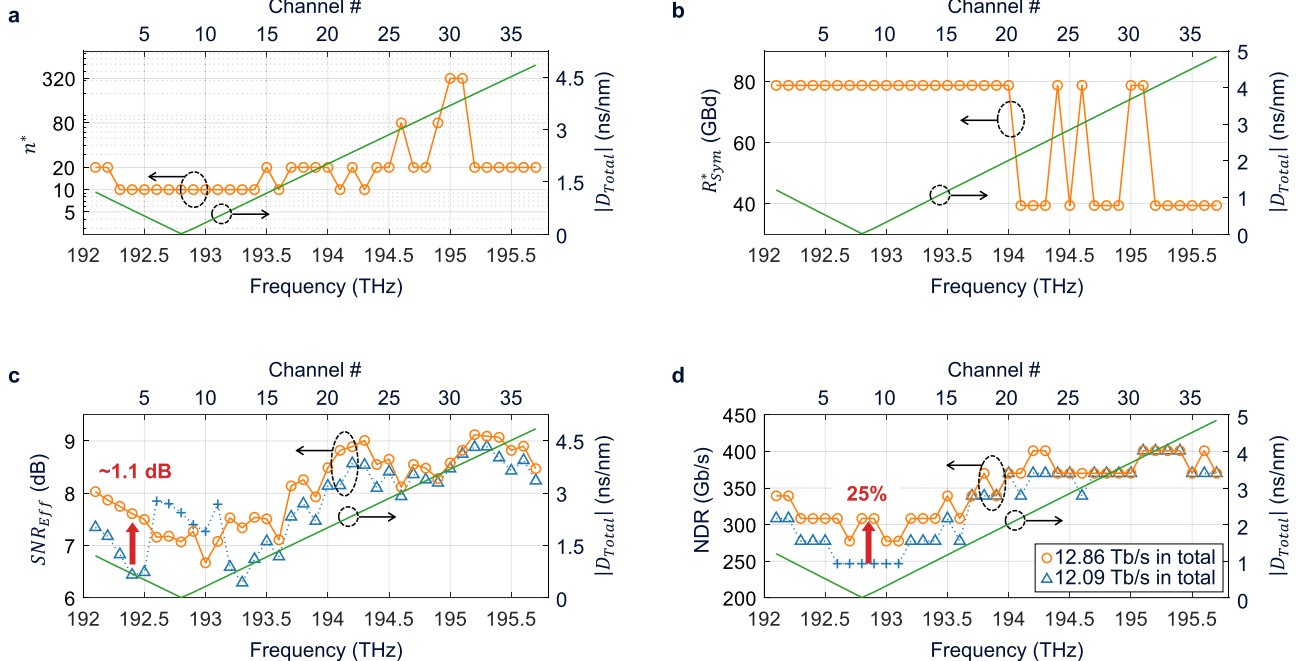

**Fig. 5 Experimental results obtained with the nonlinear fiber-channel-optimized sphere shaping. a** Optimal shaping length $n^*$ for maximum NDR that tends to increase with the total net dispersion $|D_{Total}|$. **b** Optimal symbol rate $R^*_{Sym}$ for maximum NDR that decreases from 78.8 to 39.4 GBd when $|D_{Total}|$ increases from approximately 2 to 4 ns/nm. **c** The effective SNR achieved by the fiber-channel-optimized $n^*$ and $R^*_{Sym}$ (orange circles) and by the linear-channel-optimized $n = 320$ at $R_{Sym} = 39.4$ GBd (blue triangles and pluses). **d** The NDRs achieved by the fiber-channel- optimized and linear-channel-optimized sphere shaping configurations. Near the zero-dispersion regime, an NDR increase of up to 25% is achieved by the fiber-channel-optimized shaping.

reasons that explain the phenomenon. While Eqs. (7) and (8) are valid for all system configurations tested in this article, it remains for future work to see if the optimal measurement window size changes in different system configurations, such as when the total system bandwidth changes. It also remains for future work to see how the dependence of $\eta$ on $R_{Sym}$ and $n$ changes as the sphere-shaped QAM modulation order increases to approach continuous Gaussian signaling in terms of the time-averaged probability distribution.

The finding of this article that there is a specific time window for measuring power deviations that allows the manifestation of Kerr nonlinearity to be most accurately envisioned, has significant practical implications for optical fiber communications. For the sphere shaping demonstrated in this article, identifying such an optimal time window allows us to analytically determine the optimal spectro-temporal sphere shaping configuration for maximum communication performance. It also opens up the possibility of finding new modulation techniques for lightwaves that minimize power deviations using the exact measurement window, which will be different from what are incorrectly found using a single symbol window.

It should be noted that the simulation and experiment presented in this paper focus on the study of the light propagation effects by excluding the impact of imperfect transceivers. In real world applications, however, determining the system parameters is a much more complex problem since the implementation cost and technological limitations must be considered. For example, in the presence of the spectral roll-off of high-bandwidth transceiver components, or due to the complexities associated with multi-carrier DSP, single-carrier transmission at the maximum band-width allowed by the technology can find an advantage over DSM. Also, the use of advanced carrier recovery algorithms such as the maximum-likelihood blind phase search (BPS) may influence the impact of sphere shaping on NLI under certain

conditions[47], but in this work at transmission distances that match the sphere-shaped 16-QAM format, no noticeable effect was observed using the BPS.

## Methods

**Sphere shaping.** The digital sphere shaping encoder is implemented by enumerative sphere shaping (ESS)[7,8] for $n = 5, 10, 20, 40, 80$, and by constant composition distribution matching (CCDM)[9,10] for $n = 320, 1280, 5120$. Note that CCDM can only approximately realize sphere shaping for finite block lengths, but it converges to ideal sphere shaping with a decreasing approximation error as the block length increases. The sphere shaping encoder produces $4n$-long amplitude blocks to conform with the amplitude-to-PDM symbol mapping rule used in the article. For the case of $R = 6.4$ bits per dual-polarization symbol, the finite-length shaping blocks used in the split-step simulation and C-band transmission experiment consume 0.583, 0.346, 0.209, 0.137, 0.099, 0.025, 0.009, 0.002 dB more average symbol energy than ideal shaping, respectively, for $n = 5, 10, 20, 40, 80, 320, 1280, 5120$. Here, the ideal shaping refers to an i.i.d. process that creates dual-polarization symbols according to a Maxwell-Boltzmann distribution over the given support at the same entropy rate as $R$.

**Split-step simulation.** The fiber loss parameters are $\alpha_{SSMF} = 0.2$ dB/km and $\alpha_{DCF} = 0.45$ dB/km, where the subscripts denote the fiber type. The fiber dispersion parameters are $\beta_{2,SSMF} \approx -2.199 \times 10^{-26}$ s²/m and $\beta_{2,DCF} \approx 1.020 \times 10^{-25}$ s²/m. The fiber nonlinearity parameters are $\gamma_{SSMF} = 1.45 \times 10^{-3}$ W⁻¹/m and $\gamma_{DCF} = 5.02 \times 10^{-3}$ W⁻¹/m. All fiber parameters are quantified at a wavelength of 1550.1 nm. We transmit $2^{18}, 2^{17}, 2^{16}, 2^{15}, 2^{14}, 2^{14}, 2^{14}$ dual-polarization symbols in each channel, respectively, for the WDM systems with $N_{Ch} = 1, 2, 4, 8, 16, 32, 64$. The total number of dual-polarization symbols transmitted over *all* channels is therefore $2^{18}$ for $N_{Ch} \le 16$, $2^{19}$ for $N_{Ch} = 32$, and $2^{20}$ for $N_{Ch} = 64$. The modulated symbols are oversampled with factors of 2, 4, 8, 16, 32, 64, 128, respectively, for $N_{Ch} = 1, 2, 4, 8, 16, 32, 64$; i.e., the oversampling factor increases linearly with the symbol period $T_{Sym}$. The step size is adaptively determined by the distance at which the nonlinear phase shift in a single step is 0.1 degrees, or 10 km, whichever is smaller. Every EDFA along the optical fiber links has a noise figure of 4.5 dB and produces a constant output power equal to the launch power such that the ASE-induced signal droop is accurately evaluated[48]. For all system configurations, we calculate $SNR_{Eff}$ by dividing the total signal power within the entire 100 GHz-wide frequency band by the total ASE plus NLI power within the same band. While the experiment is performed with full C-band transmission for practical significance,

the split-step simulation is performed over a total bandwidth of 100 GHz due to the simulation time required for the many configurations under test; seven symbol rates, eight shaping block lengths, six launch powers over four different links correspond to 1344 simulation runs.

**EGN simulation**. The second order and fourth order nonlinear coefficients of the EGN model[30] are obtained with Monte-Carlo integration over $10^6$ random realizations of frequency tones in the system bandwidth, for each of the SPM and XPM, using the same sets of system parameters as the split-step simulation. From the windowed central moments of $\not{p}$, the fourth and sixth standardized moments of $x$ can be obtained as $\bar{\mu}_2 + 1$ and $\bar{\mu}_3 + 3\bar{\mu}_2 + 1$, respectively, and then substituted into the EGN model. Due to the different optimal window sizes $w$ for SPM and XPM, as shown in Eqs. (7) and (8), different standardized moments are used to evaluate SPM and XPM for each set of system parameters.

**Full C-band transmission experiment**. The laser for the TX at CUT is a semiconductor external cavity laser (ECL). The ECL's wavelength is fully tunable in C-band, typical output power is 13 dBm, and linewidth is 40 kHz. The same type of laser is used as the local oscillator (LO) of the coherent RX. For the other TXs for 36 interfering channels, distributed feedback (DFB) lasers are used. The DFB lasers' typical output power is 8 dBm and linewidth is 10 MHz. The modulators are LiNbO₃ modulators with 3 dB bandwidth of 35 GHz. The half-wave voltage ($V_\pi$) of the modulator is 3 V. The lightwaves from the 36 DFB lasers are multiplexed into even and odd frequency bands using polarization maintaining silica based arrayed waveguide grating multiplexers (AWGs). We modulate the lightwaves with dual-polarization I/Q modulators for all channels. At the TX, $336896/N_{SC}$ sphere-shaped dual-polarization symbols are generated by using a computer for each of $N_{SC}$ subcarriers. To assist coherent DSP of the RX, one QPSK symbol is inserted as a pilot for every 47 sphere-shaped symbols, using the same average power as the sphere-shaped symbols. The number of modulation symbols is limited by the memory size of the digital-to-analog converters (DACs). The generated $N_{SC}$ symbol streams are filtered by RRC filters with a pass bandwidth of $R_{Sym}/N_{SC}$ GHz and roll-off factors between 0.05 and 0.1, digitally frequency-shifted and multiplexed to build a comb of $N_{SC}$ digital subcarriers in the baseband. Then, the combined $N_{SC}$ symbol streams are filtered by another RRC filter with a pass bandwidth of $R_{Sym}$ GHz and a roll-off factor of 0.1. Digital pre-distortion[49] in the frequency domain follows and compensates for the bandwidth limitations of the TX components such that the optical signal launched into fiber has a flat power spectral density (PSD) in the pass band. Four 17 nm CMOS DACs with 8-bit resolution transform the generated digital symbol streams to electrical fields at a sampling rate of 120 GSa/s. Each subcarrier is modulated at a symbol rate of $R_{Sym}/N_{SC} = 9.84375, 19.6875, 39.375, 78.75$ GBd, respectively, for $N_{SC} = 8, 4, 2, 1$. The maximum symbol rate is determined by $R_{Sym} = 78.75$ GBd such that DSM (with $N_{SC} \geq 2$) can perform similar to $N_{SC} = 1$ in a back-to-back configuration, since at higher symbol rates the DSM performs worse than $N_{SC} = 1$ due to the bandwidth limitations of the transmitter components. The oversampling factors correspond to $120/R_{Sym} \approx 12.19, 6.10, 3.05, 1.52$, respectively, for $N_{SC} = 8, 4, 2, 1$. At the TX, interfering channels are split by a liquid crystal on silicon (LCoS)-based wavelength selective switch (WSS), propagate through different delay fibers for decorrelation with each other, and recombined by another WSS of the same type.

In the recirculating loop, EDFAs with a noise figure around 4.7 dB are used. A variable optical attenuator (VOA) is attached after each EDFA, such that the optical power at the output of the VOA equals the launch power. The PSD of the optical signal at the EDFA output is flattened by an LCoS-based or micro-electromechanical systems (MEMS)-based programmable optical filter that acts as a DGE.

At the RX, an LCoS-based WSS filters out interfering channels, leaving only the CUT. Four balanced photo diodes (BPDs) with 75 GHz bandwidth detect the optical signals on the CUT. The electrical signals produced by the BPDs, containing $N_{SC}$ symbol streams inside, are sampled at 256 GSa/s each by four analog-to-digital converters (ADCs) in a real-time oscilloscope. The digitized waveforms are resampled at 157.5 GSa/s, which will eventually lead to an oversampling factor of 2 for any $N_{SC}$. Chromatic dispersion is digitally compensated for in the combined $N_{SC}$ symbol streams. Each of the $N_{SC}$ symbol streams is then frequency-shifted to baseband, match-filtered by an RRC filter with a pass bandwidth of $R_{Sym}/N_{SC}$ GHz, and down-sampled to 2 samples per symbol. For each separated symbol stream, after recovery of the symbol timing[50], real-valued PDM equalization[51] is performed in the frequency domain with carrier phase recovery based solely on pilot symbols. We calculate $SNR_{Eff}$ in Fig. 5(c) by dividing the total signal power for $N_{SC}$ symbol streams by the total ASE plus NLI power for $N_{SC}$ symbol streams. Likewise, the NDR in Fig. 5(c) is calculated on an optical carrier basis.

## Data availability
The raw data underlying Figs. 2 and 3(a) are available at https://figshare.com/articles/dataset/Shaping_Lightwaves_in_Time_and_Frequency/17029454. The raw data underlying all other figures can be reproduced easily using a computer, or their approximate values may be readily obtained from the figures, and are provided from the corresponding author upon reasonable request.

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

## Author contributions

J.C. proposed the idea. J.C. and S.O. performed the numerical simulations. J.C., X.C., G.R., D.C., and E.B. performed the experiments. J.C., X.C., G.R., D.C., E.B., S.O., and R.T. discussed the idea and analyzed the results. R.T. supervised the project. J.C. wrote the draft. X.C., G.R., D.C., S.O., and R.T. substantially improved the paper.

## Competing interests

The authors declare no competing interests.
