## [Peer Review File · Nature Communications]

Review of NCOMMS-21-15776: “Shaping Lightwaves in Time and Frequency for Optical Fiber Communication”

The authors of the paper investigate constellation shaping (sphere shaping in particular) for optical communication systems. They focus on the effect of (1) the block length of sphere shaping and (2) the symbol rate of the signaling on the amount of nonlinear interference (NLI) generated during propagation. The dependency of NLI on the shaping block length has been investigated in the literature. The authors propose a new metric (windowed central moment) to study this dependency. This metric measures the high-order deviations of the energy of the optical field from its average over a finite time interval. They claim that properly selecting the width of this window is important to properly compare different shaping schemes from NLI generation perspective and to predict their performance. Furthermore, they claim that the nonlinearities depend on the energy structure of the optical field in absolute time rather than in number of symbols. This leads to a complex relation between the shaping block length, symbol rate and NLI. Using the values of the window length at which the correlation between the windowed moment and the effective SNR is maximized, authors claim that the EGN model predicts the effective SNR (using windowed moments) more accurately than its traditional form (using regular moments) (Fig. 3(b)).

The paper is well-written, findings are interesting, the topic is timely/relevant, and the explanation of the experimental setup is very clear and detailed. I believe the paper can be accepted after a round of revision/polishing. The structure of the paper is different than I am used to from other journals in the literature and this structure may be leading to a different reading experience than I am used to. However I must say that the introduction can be improved by stating (1) the problem, (2) what is done, and (3) what is found more clearly with short and direct sentences. At some points, I had trouble following the train of reasoning. I believe the findings of the paper are quite interesting, and they should be stated more clearly to increase their impact. In the following, I will list my comments which I hope would help the authors to improve the presentation in the paper.

- Firstly, I find it a bit complicated to understand the fundamental motivation and findings of the paper from the abstract and the introduction. I suggest a clearer statement of the fact that (1) the paper investigates the optimal combination of the shaping block length and the symbol rate that maximizes the performance, (2) this is achieved using a new metric—similar to a metric proposed in [1]—that has high correlation with the effective SNR. At some parts of the paper, it is implied that a NLI-optimized sphere shaping approach is proposed. However, as far as I understand, only the optimum system parameters are investigated (which is also a valuable contribution) for regular sphere shaping.
- In the traditional EGN model, higher order moments of the channel input distribution are used in the expressions that compute the NLI variance. Here, they are replaced with their windowed versions and the reported results match the actual performance more accurately than the original EGN model. Can the authors comment on the use of the windowed moments in the EGN model? Is there a theoretical background for this, or is it just heuristic? Can this approach be used to propose an enhanced EGN model that works for finite-length shaping schemes and captures the temporal structure of the shaped waveforms in the analytical model?
- The improvement in rate (or in reach) due to shaping (and sphere shaping in particular) has been demonstrated in the literature repeatedly. The paper provides an optimization of shaping over (i) block length and (ii) symbol rate. There are also other works in the literature which discuss the effect of block length and symbol rate on performance. They should at least be cited, e.g., [2], [3], [4], [5], etc.
- In p.5, it is stated that refs. 7 and 15 show that if n is small as in Fig. 1(c), sphere shaping reduces NLI. Firstly, ref. 15 is not about sphere shaping, but about constant composition sequences. Secondly, as previously observed in [6] and re-stated in ref. 7, (AWGN-optimal) shaping leads to increased NLI (decreased effective SNR). This is attributed to the increased kurtosis in [6], [7]. Therefore, I am not sure I see the contradiction that you mention here. Furthermore, the explanation concerning Fig. 1(d) is not clear. Are we focusing on the effect of n on NLI here or the importance of selecting w to predict the performance? Because regardless of w , smaller n implies smaller $\bar{\mu}_2$ in Fig. 1(d) (for most of the region). I would suggest a careful revision of the paragraph in p.5 that starts with “Fig. 1(c)” (which I believe should be “Figure 1(c)”) with clear explanations (of the claims and figures) and as much connections to the existing literature as possible. I believe this paragraph is extremely important to motivate your work.
- As far as I know, the effective SNR in EGN model depends on the average power of the input as well as its regular fourth and sixth order moments. In p.8, how do you use the EGN model with $\bar{\mu}_2$ and $\bar{\mu}_3$ to compute the effective SNR? Figure 3(b) reports that this computation (EGN with windowed moments) is far more accurate than the regular EGN model in predicting the effective SNR. If this is the case, would you claim that you’re proposing an enhanced EGN model?

Furthermore, is it possible to obtain a further more accurate relation if (somehow) some other higher-order windowed moments are included in the model? Also, when you say “optimized $\bar{\mu}_2$ and $\bar{\mu}_3$ ” here, do you mean their values computed with the optimum window length?

In the following, I list some minor comments.

- 1) p.2 paragraph 1: “ever-increasing demand for communication capacity” \rightarrow “ever-increasing demand for higher data rates”
- 2) p.2 paragraph 2: Refs. 8 and 9 are given as references for sphere shaping, but they are not. I would suggest instead [8], [7].
- 3) p.2 paragraph 2: “block of amplitudes” is rather vague, it should be related to “communication (QAM) symbols” (mentioned in the previous paragraph) better.
- 4) p.3 paragraph 2: 4 consecutive amplitudes \rightarrow one DP symbol: This is discussed extensively in [4], I would suggest referencing it.
- 5) p.3: $\|\cdot\|^2$ is not defined
- 6) p.3: Explanation of sphere shaping can be connected to the related literature more strongly
- 7) p.3 paragraph 1: “(...) if every shaped block is plotted as a point in $4n$ -dimensional signal space, with the i -th amplitude being the position of the point on the i -th coordinate axis, the points uniformly fill a $4n$ -dimensional (hyper-)sphere (...)” \rightarrow Since the amplitudes are drawn from a discrete set, this is not correct. The points are located in a $4n$ -dimensional spherical region of a $4n$ -dimensional rectangular lattice. I would avoid the use of the phrase “uniformly fill”.
- 8) p.4: I would recommend the use of R instead of H for the information rate, since H is almost always used to denote the entropy
- 9) p.4 Fig. 1(a): Sphere shaping indeed decreases the maximum energy substantially. However, what is more important is the decrease in average energy. I would recommend indicating the decrease in average energy, and even relating it to the “linear” shaping gain discussed, e.g., in [9].
- 10) p.4 Fig. 1(b): I do not understand what this figure tries to explain, or how Fig. 1(a) relates to this.
- 11) p.4: sentence before (1) and (2): “(...) η increases as the central moment (...)” $\rightarrow \eta$ or P_{NLI} ?
- 12) p.5: “(...) deviation by the instantaneous power (...)” \rightarrow “(...) deviation of the instantaneous power (...) from average $\langle p \rangle = 1$ (...)” ?
- 13) p.5: “(...) appears to be more spread around the average (...)” \rightarrow I think this can be related to the difference in their $\bar{\mu}_2$, right? If so, please refer to these values.
- 14) Figures 1(c-d): What is the shaping rate and the resulting distribution?
- 15) p.6: I do not understand what “With our system settings, $n = 5120$ approximately represents i.i.d. shaping from the dispersion perspective, since each symbol is dispersed over less than 5120 symbol periods in all conditions except for $R_{Sym} = 88$ GBd in Link D.” exactly means.
- 16) p.6: I am not sure how the observation (iii) is explained with “(...) since increasing R_{Sym} reduces the shaping block duration in absolute time for the same n ”.
- 17) p.7: I understand the observation “Areas far from the circled areas have a negligible impact on SNR_{Eff} . But I believe this can be related to Figs. 2(a,c) more strongly and quantitatively.
- 18) Fig. 3(a): How many spans?
- 19) p.8: You say “optimal sphere shaping” more than once in the paper. What exactly do you mean by “optimal”? The combination of n and R_{Sym} that maximizes the performance? There is an optimal window length w that maximizes the correlation b/w the windowed moment and the effective SNR. But I am not sure what you mean by “optimal” sphere shaping. Clarification is needed.
- 20) p.9: I believe the conclusion “In general, n^* increases as $|D_{Total}|$ (green solid line) increases, (...)” is a bit strong to be made from Fig. 5(a). The optimal block length here is mostly constant around 10-20, increasing around 195 THz, but then decreasing again. If the authors can provide some additional explanation for this behaviour, it would make the paper stronger.
- 21) p.9: At which block length Fig. 5(b) is plotted? Also, similar to my previous comment, the conclusion “(...) R_{Sym}^* decreases (...) when $|D_{Total}|$ increases (...)” is a bit strong to be made from Fig. 5(b). Instead of simply saying Figs. 5(a,b) agree with Figs. 2(b,a), possible explanations for the peculiar behaviours in Figs. 5(a,b) should be provided.
- 22) p.9, third line: SNR_{Eff} of Fig. 4(c), or of Fig. 5(c)?
- 23) p.11: CCDM is not a sphere shaping algorithm.
- 24) Similar gains in data rate/SNR/AIR to the ones that are reported at the end manuscript exist in the literature. A short review of such works would make the paper stronger.

REFERENCES

- [1] K. Wu, G. Liga, A. Sheikh, F. M. J. Willems, and A. Alvarado, “Temporal energy analysis of symbol sequences for fiber nonlinear interference modelling via energy dispersion index,” Feb. 2021. [Online]. Available: <https://arxiv.org/abs/2102.124114>
- [2] P. Skvortcov, I. Phillips, W. Forsyiaik, T. Koike-Akino, K. Kojima, K. Parsons, and D. S. Millar, “Nonlinearity tolerant LUT-based probabilistic shaping for extended-reach single-span links,” *IEEE Photon. Technol. Lett.*, vol. 32, no. 16, pp. 967–970, 2020.

- [3] T. Fehenberger, "On the impact of finite-length probabilistic shaping on fiber nonlinear interference," in *Proc. Signal Process. in Photon. Commun. (SPPCom)*, Washington, D.C., United States, July 2020.
- [4] P. Skvortcov, I. Phillips, W. Forysiak, T. Koike-Akino, K. Kojima, K. Parsons, and D. S. Millar, "Huffman-coded sphere shaping for extended-reach single-span links," *IEEE J. Sel. Topics Quantum Electron.*, vol. 27, no. 3: 3500215, May-June 2021.
- [5] S. Civelli, E. Forestieri, and M. Secondini, "Interplay of probabilistic shaping and carrier phase recovery for nonlinearity mitigation," in *Proc. Eur. Conf. Opt. Commun. (ECOC)*, Brussels, Belgium, Dec. 2020.
- [6] T. Fehenberger, A. Alvarado, G. Böcherer, and N. Hanik, "On probabilistic shaping of quadrature amplitude modulation for the nonlinear fiber channel," *J. Lightw. Technol.*, vol. 34, no. 21, pp. 5063–5073, Nov. 2016.
- [7] Y. C. Gültekin, A. Alvarado, O. Vassilieva, I. Kim, P. Palacharla, C. M. Okonkwo, and F. M. Willems, "Kurtosis-limited sphere shaping for nonlinear interference noise reduction in optical channels," June 2021. [Online]. Available: <https://arxiv.org/abs/2105.14794>
- [8] P. Schulte and F. Steiner, "Divergence-optimal fixed-to-fixed length distribution matching with shell mapping," *IEEE Wireless Commun. Lett.*, vol. 8, no. 2, pp. 620–623, Apr. 2019.
- [9] Y. C. Gültekin, W. J. van Houtum, A. Koppelaar, and F. M. J. Willems, "Enumerative sphere shaping for wireless communications with short packets," *IEEE Trans. Wireless Commun.*, vol. 19, no. 2, pp. 1098–1112, Feb. 2020.

Reviewer #2 (Remarks to the Author):

This work studies the optimization of the time and bandwidth over which sphere shaping should be performed to minimize the impact of nonlinear effects and maximize performance in optical fiber communication.

The work is timely, well written, and reports some novel interesting results.

The optimization of the shaping block length in the nonlinear regime has been already investigated in some other papers but, to my knowledge, the simultaneous optimization of block length and symbol rate (i.e., time and bandwidth) is studied here for the first time. Moreover, this study encompasses several scenarios and a wide range of parameters, both numerically and experimentally.

The methodology is sound and the methods are described with enough detail. Moreover, the conclusions are supported both by numerical and experimental results.

In general, I believe that the paper deserves publication. However, there are some weaknesses and some issues that should be considered and possibly addressed, as detailed below.

1) As a general note, while reading the paper, I was expecting to find a theoretical analysis which explains and predicts the results observed numerically and experimentally. This expectation is not fully met by the paper, which provides only some insight about the observed phenomena and some empirical laws to model them - e.g., (7), (8) and the use of the windowed moments in the EGN model. In fact, there are a number of perturbation models in the literature which could be used to attempt such an analysis, for instance by removing the i.i.d. assumption in the EGN model. An example can be found in Liga et al "Extending Fibre Nonlinear Interference Power Modelling to Account for General Dual-Polarisation 4D Modulation Formats", Entropy, 2020. I recommend discussing this issue and highlighting the empirical nature of the proposed model.

2) As a second general comment, the analysis practically neglects the impact of carrier recovery. Indeed, this might be substantial, as a sufficiently fast carrier recovery algorithm can mitigate the impact of nonlinear phase noise and change the dependence of performance on block length (see for instance Civelli et al. "Interplay of probabilistic shaping and carrier phase recovery for nonlinearity mitigation", ECOC 2020). With carrier recovery, there might be no advantage in using short (optimized) block lengths. It seems that carrier recovery is not present in the simulations and is quite slow in the experiments (using only 1 pilot every 48 symbols), possibly overestimating the advantage of using a short optimized block length. Please comment about that. As a side note, for the sake of transparency, I am a coauthor of the above mentioned paper. I do not want to push my own work, therefore feel free to decide if the paper is relevant or not to your work and if it deserves to be cited or not. This decision will not affect my final recommendation.

3) Abstract: The sentence "In optical fiber, however, the sphere shaping induces Kerr nonlinearity in a peculiar way that makes analysis of transmission performance difficult, potentially lowering the communications capacity" sounds a bit odd. It suggests that it is sphere shaping that makes the

analysis difficult, with Kerr nonlinearity just playing an indirect role. In fact, I think the opposite is true. Moreover, it suggests that the difficulty of the analysis may play a role in the capacity reduction.

4) Introduction: The idea of using sphere shaping on an optimized block length to minimize the impact of nonlinear interference was first proposed in [10] and then further investigated by some of the same authors in Geller et al. "A Shaping Algorithm for Mitigating Inter-Channel Nonlinear Phase-Noise in Nonlinear Fiber Systems", JLT 2016. Your work (as well as some other recent works) heavily relies on this idea, extending the optimization to the frequency domain. I think that it would be fair to better acknowledge this fact and the role of these two papers.

5) Page 3, Results: you state that "The symmetry of the probability allows for legitimate analysis with only positive amplitudes, and hence we omit the sign throughout this article". I guess that the presence of a random sign with uniform distribution is implicitly assumed in the paper and neglected only in the description, whereas it is included both in the simulations and in the experiments. Is that correct? I am not sure that transmitting only positive amplitudes would indeed give the same results. I think that two consecutive symbols with the same sign or with opposite signs induce a fundamentally different response in the nonlinear channel. Please be more explicit.

6) Page 3, Results: the sentence "one dual-polarization symbol that maintains the value over a period of T_{sym} (s) and can change the value at a rate of $R_{\text{sym}}=1/T_{\text{sym}}$ (Bd)", suggests a rectangular pulse shape. I suggest revising the sentence. Moreover, the indication of the units of measure in this way is neither standard nor required.

7) Page 5, 4 lines before the end of the subsection: The sentence "As w increases, μ_2 remains the same for all types of i.i.d. symbols" is misleading. It is not "the same" for all types of symbols but, rather, it remains "constant" as w increases for a given modulation (type of symbols).

8) Equation (3): for sphere shaping, the transmitted signal is a cyclostationary process with period equal to the length of the shaped blocks (unless a uniform random delay is assumed). Therefore, I guess that the windowed moments depend on the time at which the window is centered, unless they are averaged over it. Please comment about that and explain if and how you have considered this time dependence in your analysis.

9) Page 5, Section "Optimization of Sphere Shaping in the Time-Frequency Plane": Please specify what QAM format is considered in this section. I guess from the given data that it is 16QAM (as in the previous section). Yet, an explicit mention here would be useful.

10) Page 6: You state that "With our system settings, $n=5120$ approximately represents i.i.d. shaping from the dispersion perspective, since each symbol is dispersed over less than 5120 symbol periods

in all conditions except for $R_{\text{sym}}=88$ GBd in Link D" and that "like the i.i.d. uniform QAM, the optimal R_{sym} for i.i.d. shaping to maximize SNR_{eff} (yellow stars) decreases as the total net dispersion D_{Total} increases". I think that this discussion is a bit weak and that the analogy with the symbol rate optimization for uniform i.i.d. QAM is misleading for two reasons: 1) The dependence of SNR_{eff} on symbol rate for the uniform i.i.d. QAM case vanishes for Gaussian symbols (the EGN and GN model converge to the same equations, see for instance [23]). Therefore, I expect a very little dependence on the symbol rate for shaped i.i.d. symbols, as they approximate i.i.d. Gaussian symbols. In fact, the dependence on symbol rate in Fig. 2a for the case $n=5120$ is very weak (and would be probably even weaker for shaped 64-QAM or 256-QAM) and the peak SNR is not so different from the surrounding values in the simulation range; 2) assuming that $n=5120$ represents the i.i.d. case from a dispersion perspective is an approximation whose accuracy decreases as the symbol rate increases; therefore, it cannot be used when explaining the (weak) dependence of SNR_{eff} on symbol rate. In fact, for $n=5120$, this dependence becomes a little more relevant exactly where the approximation becomes looser. In my opinion, it seems more likely that the dependence of SNR_{eff} on symbol rate is mainly explained, even for $n=5120$, by the same reason that explains it for lower n .

11) Page 6, you state that "the optimal R_{sym} is higher for finite-length shaping than for i.i.d. shaping (compare the yellow and red stars), since increasing R_{sym} reduces the shaping block duration in absolute time for the same n ". Following up on the previous comment, can we better (and more simply) say that the optimal symbol rate increases when n decreases?

12) Still on the same matter: you refer many times in the results to the i.i.d. case, considering $n=5120$ as an approximation to it. Why not directly simulating the true i.i.d. case? It should be sufficient to draw i.i.d. QAM symbols from a Maxwell-Boltzmann distribution. That would be a more meaningful benchmark against which to compare the results for finite block lengths and, for instance, to compute η^{∞} in (6).

13) Page 8, Demonstration of Optimal Sphere Shaping through Full C-Band Transmission Experiment: The experiment is performed on a link that is substantially different from the 4 links considered in the simulation. Why? In particular, each channel sees a different dispersion profile, making the interpretation of the results harder. Moreover, the important dispersion-unmanaged case is not considered in the experiments. Is there a particular reason for this choice? Is it instrumental to highlight some effect or behavior? Please explain and motivate this choice.

14) Page 11, Methods: you state that "The digital sphere shaping encoder is implemented by enumerative sphere shaping (ESS) for $n=5, 10, 20, 40, 80$, and by constant composition distribution matching (CCDM) for $N=320, 1280, 5120$." and "...consume 0.583, 0.346, 0.209, 0.137, 0.099, 0.025, 0.009, 0.002 dB more average symbol energy than ideal shaping, respectively". Strictly speaking, CCDM does not implement sphere shaping (as all constant composition sequences lie on the surface of the sphere, partially covering it) and, in fact, its energy loss is higher than that of ESS for the same blocklength. Of course, they both asymptotically converge to i.i.d. Maxwell-Boltzmann symbols, but as you are studying how performance changes with block length, you cannot assume that such an

asymptotic regime has been achieved. In fact, I am also a bit surprised that CCDM with $n=320$ performs significantly better than ESS with $n=80$ (see, e.g., the comparison in [7]). Please comment about that and double check the numbers (I have not verified them, it is just a feeling and I might be wrong).

Marco Secondini

Response to Reviewers' Comments on NCOMM-21-15776 "Shaping Lightwaves in Time and Frequency for Optical Fiber Communication"

Junho Cho, Xi Chen, Greg Raybon, Di Che, Ellsworth Burrows, Samuel Olsson, and Robert Tkach

Dear Reviewers,

We sincerely thank all reviewers for carefully reading and commenting on the paper. We addressed in the revised manuscript all the suggestions and comments, which we believe has significantly improved the paper. The changes made in the revised manuscript are **highlighted in yellow**. Please find our point-by-point response to the comments below, where the reviewers' comments are reproduced verbatim in *blue italic type*, and our response is written in black roman type. Note that the figure numbers, table numbers and reference numbers below indicate those of the revised manuscript, unless specified otherwise.

Response to Reviewer 1:

The authors of the paper investigate constellation shaping (sphere shaping in particular) for optical communication systems. They focus on the effect of (1) the block length of sphere shaping and (2) the symbol rate of the signaling on the amount of nonlinear interference (NLI) generated during propagation. The dependency of NLI on the shaping block length has been investigated in the literature. The authors propose a new metric (windowed central moment) to study this dependency. This metric measures the high-order deviations of the energy of the optical field from its average over a finite time interval. They claim that properly selecting the width of this window is important to properly compare different shaping schemes from NLI generation perspective and to predict their performance. Furthermore, they claim that the nonlinearities depend on the energy structure of the optical field in absolute time rather than in number of symbols. This leads to a complex relation between the shaping block length, symbol rate and NLI. Using the values of the window length at which the correlation between the windowed moment and the effective SNR is maximized, authors claim that the EGN model predicts the effective SNR (using windowed moments) more accurately than its traditional form (using regular moments) (Fig. 3(b)).

The paper is well-written, findings are interesting, the topic is timely/relevant, and the explanation of the experimental setup is very clear and detailed. I believe the paper can be accepted after a round of revision/polishing. The structure of the paper is different than I am used to from other journals in the literature and this structure may be leading to a different reading experience than I am used to. However I must say that the introduction can be improved by stating (1) the problem, (2) what is done, and (3) what is found more clearly with short and direct sentences. At some points, I had trouble following the train of reasoning. I believe the findings of the paper are quite interesting, and they should be stated more clearly to increase their impact. In the following, I will list my comments which I hope would help the authors to improve the presentation in the paper.

Response: Thank you for all the constructive comments here and in what follows. We have tried to address your concerns by improving the presentation as much as we can. We hope that the ambiguities found in the previous manuscript have completely been removed in the revised manuscript and the content of the paper is now much clearer.

Firstly, I find it a bit complicated to understand the fundamental motivation and findings of the paper from the abstract and the introduction. I suggest a clearer statement of the fact that (1) the paper investigates the optimal combination of the shaping block length and the symbol rate that maximizes the performance, (2) this is achieved using a new metric—similar to a metric proposed in [1]—that has high correlation with the effective SNR. At some parts of the paper, it is implied that a NLI-optimized sphere shaping approach is proposed. However, as far as I understand, only the optimum system parameters are investigated (which is also a valuable contribution) for regular sphere shaping.

Response: As the reviewer suggested, we made the motivation and contribution of this paper clearer in the abstract and the introduction. The changed sentence in the abstract is “**In this article, we show that the impact of sphere shaping on Kerr nonlinearity varies with chromatic dispersion, shaping block length and symbol rate, and that this impact can be predicted using a novel statistical measure of light energy.**” In the abstract, a longer explanation on the motivation and contribution of this work was avoided to meet the suggested length of 150 words, but we tried to make it as clear as possible within the given length. In the introduction, several sentences have been changed to clarify the motivation and contribution of this paper, as highlighted in yellow in the upper half of Page 3. In particular, we emphasized that the previous studies have focused only on the block length optimization, neglecting the influence of the symbol rate and chromatic dispersion. It is only by knowing the findings of this work that the results can be consistently explained between several independently performed experiments that used different settings. To clarify the contribution of this work, we added “**While the previous works^{8,11,12,16,23} optimized only n to observe some gains in SNR_{Eff} and NDR over specific links (e.g., for single-span links), joint optimization of n and R_{Sym} in this work produces significantly larger gains and allows these gains to be achieved over a much wider variety of links.**” at the end of the “Demonstration of Optimal Sphere Shaping through Full C-Band Transmission Experiment” section. We also made it clear throughout the abstract, introduction, and main text that it is not the shaping algorithm that we optimize to minimize the NLI, but the parameters of the regular sphere shaping.

In the traditional EGN model, higher order moments of the channel input distribution are used in the expressions that compute the NLI variance. Here, they are replaced with their windowed versions and the reported results match the actual performance more accurately than the original EGN model. Can the authors comment on the use of the windowed moments in the EGN model? Is there a theoretical background for this, or is it just heuristic? Can this approach be used to propose an enhanced EGN model that works for finite-length shaping schemes and captures the temporal structure of the shaped waveforms in the analytical model?

Response: Thanks for this valuable comment. At a recent conference, we learned from an author of Ref. 26 that an extended EGN model was published last year to deal with structures (or correlations) present in four amplitudes (i.e., in one symbol period). A mathematically unequivocal approach to modeling the propagation of structured lightwaves would be to further extend Ref. 26 (see also Ref. 27)

for structures spanning more than one symbol period. However, given the enormous complexity of the mathematical expansion to account for only four correlated amplitudes, it can easily be inferred that it is mathematically daunting to find an accurate analytical model to handle tens to thousands of correlated amplitudes that we cover in this paper. Also, as seen from Refs. 26-27, the resulting equations of the extended EGN model for many correlated amplitudes may have too many complex terms to be practically useful.

In this work, we allowed for model inaccuracies caused by the assumptions on i.i.d. amplitudes, and attempted to solve the problem with manageable complexity by substituting the windowed moments that can characterize structured lightwaves statistically and quantitatively into the traditional EGN model, whether the structure is short or long. We have clarified this in the revised manuscript with properly added citations, as found in the first paragraph of Page 9. It may be possible to apply our approach to the extended EGN model of Ref. 26 to see if the prediction accuracy is even more improved, which we would like to leave for future work.

The improvement in rate (or in reach) due to shaping (and sphere shaping in particular) has been demonstrated in the literature repeatedly. The paper provides an optimization of shaping over (i) block length and (ii) symbol rate. There are also other works in the literature which discuss the effect of block length and symbol rate on performance. They should at least be cited, e.g., [2], [3], [4], [5], etc.

Response: We thank you for letting us reference the missing important prior studies. By citing Refs. 15-17, 23 (which are [2]–[5] in your reference numbers) at the end of the second and third paragraphs of the revised introduction, we noted that there are several recent works addressing the effect of the shaping block length. However, those papers are cited only in the context of block length, since there are no existing studies on the effect of the symbol rate on Kerr nonlinearity. For example, in Refs. 8, 11, 15-17, only single symbol rates of 45 GBd, 100 GBd, 56 GBd, 56 GBd, and 50 GBd are used, respectively. In Ref. 23, signal is modulated with two different symbol rates of 42 GBd and 64 GBd, but from this no systematic analysis of the effect of symbol rate can be made. We therefore cited the prior studies in the context of the shaping block length only.

In p.5, it is stated that refs. 7 and 15 show that if n is small as in Fig. 1(c), sphere shaping reduces NLI. Firstly, ref. 15 is not about sphere shaping, but about constant composition sequences. Secondly, as previously observed in [6] and re-stated in ref. 7, (AWGN-optimal) shaping leads to increased NLI (decreased effective SNR). This is attributed to the increased kurtosis in [6], [7]. Therefore, I am not sure I see the contradiction that you mention here. Furthermore, the explanation concerning Fig. 1(d) is not clear. Are we focusing on the effect of n on NLI here or the importance of selecting w to predict the performance? Because regardless of w , smaller n implies smaller $\bar{\mu}_2$ in Fig. 1(d) (for most of the region). I would suggest a careful revision of the paragraph in p.5 that starts with “Fig. 1(c)” (which I believe should be “Figure 1(c)”) with clear explanations (of the claims and figures) and as much connections to the existing literature as possible. I believe this paragraph is extremely important to motivate your work.

Response: As the reviewer pointed out, Ref. 22 (Ref. 15 in our previous manuscript) deals with constant composition sequences. As the block length increases, the constant composition sequences can approximately realize sphere shaping with gradually decreasing rate loss, as a consequence of the sphere

hardening phenomenon. However, we admit that for short block lengths covered in Pages 5-6, constant composition sequences produce a large difference from sphere shaping. Therefore, we replaced Ref. 22 with proper citations in the sentence you referred above, and added citations to Refs. 17 and 34 (which are [6] and [7] in your reference numbers). We also added “Note that CCDM can only approximately realize sphere shaping for finite block lengths, but it converges to ideal sphere shaping with a decreasing approximation error as the block length increases” to the first paragraph in the Methods section.

As far as the “contradiction” is concerned, it is observed between the analytical model for i.i.d. symbols and the empirical results obtained using sphere shaping with short block lengths; namely, if we calculate the statistical moments of short sphere-shaped symbols with $w = 1$ as conventionally done, it is much larger than that of the unshaped symbols (see, e.g., Fig. 1(d)), and if we use the EGN model neglecting the required i.i.d. properties of symbols, these large statistical moments imply that NLI of sphere-shaped symbols is much greater than that of unshaped symbols. On the other hand, the NLI that is empirically obtained with short sphere shaping can be much smaller than that of the unshaped symbols (see, e.g., Fig. 8 of Ref. 8). In the paragraph that you mentioned, we have tried to convey the meaning of “contradiction” more clearly by rewriting several sentences, with citations to the associated prior works.

We have also paraphrased several sentences in this paragraph to avoid duplication of content or confusion between what is presented here and what is presented later sections.

Regarding the text citation of figures, some journals such as IEEE journals use the abbreviation “Fig.” even when it begins a sentence. However, for Nature Communications, we were not able to find whether “Fig. 1(c)” or “Figure 1(c)” conforms to the editorial style, so will check with the journal’s editorial team during the proofreading stage (if the paper is accepted).

As far as I know, the effective SNR in EGN model depends on the average power of the input as well as its regular fourth and sixth order moments. In p.8, how do you use the EGN model with $\bar{\mu}_2$ and $\bar{\mu}_3$ to compute the effective SNR? Figure 3(b) reports that this computation (EGN with windowed moments) is far more accurate than the regular EGN model in predicting the effective SNR. If this is the case, would you claim that you’re proposing an enhanced EGN model? Furthermore, is it possible to obtain a further more accurate relation if (somehow) some other higher-order windowed moments are included in the model? Also, when you say “optimized $\bar{\mu}_2$ and $\bar{\mu}_3$ ” here, do you mean their values computed with the optimum window length?

Response: You are correct that the EGN model requires the average power as well as the fourth and sixth standardized moments. To clarify this, we corrected a phrase in Page 9 as “Plugging the average symbol power ($\|x\|^2$) and the windowed central moments $\bar{\mu}_2$ and $\bar{\mu}_3$ obtained with w_{SPM}^* and w_{XPM}^* into a state-of-the-art analytic model known as the enhanced Gaussian noise (EGN) model”. We also added a small subsection “EGN simulation” in the Method section to explain how to convert the windowed central moments into the fourth and sixth standardized moments. We could approach mathematically as to how this conversion can be derived and why this conversion is necessary; however, we believe that this mathematical detail is beyond the scope of Nature Communications articles and that the physical rationale as addressed in the current article is more significant and relevant to this journal. Therefore, we would like to omit the mathematical details in this article.

As mentioned in an earlier response above, long mathematical derivations are needed to extend the EGN model to account for only 4 correlated amplitudes (see Ref. 26), and improving the EGN model to account for larger structures seems mathematically daunting. In this article, we are not proposing an enhanced EGN model; rather, while allowing for model mismatch by using the classical EGN model, we improve the accuracy of evaluating structured lightwaves by replacing parameters of the EGN model with improved ones (i.e., by replacing μ_2 and μ_3 with optimized $\bar{\mu}_2$ and $\bar{\mu}_3$). It is uncertain whether the existing approach to improving the model and our approach to using improved parameters can converge in the future, but regardless of the approach researchers take, we believe that our work will inspire people who have worked in this field.

Regarding your last point, you're right that "optimized $\bar{\mu}_2$ and $\bar{\mu}_3$ " means that the values of $\bar{\mu}_2$ and $\bar{\mu}_3$ are obtained with the optimum window length. We clarified this by rephrasing it as "by replacing μ_2 and μ_3 with optimized $\bar{\mu}_2$ and $\bar{\mu}_3$ (i.e., obtained with w_{SPM}^* and w_{XPM}^*)".

In the following, I list some minor comments.

1) p.2 paragraph 1: "ever-increasing demand for communication capacity" → "ever-increasing demand for higher data rates"

Response: We have modified this phrase as you suggested.

2) p.2 paragraph 2: Refs. 8 and 9 are given as references for sphere shaping, but they are not. I would suggest instead [8],[7].

Response: We believe that this paragraph should provide readers with extensive prior work on sphere shaping. Since Refs. 9 and 10 realize sphere shaping for long block lengths, we have left citations to Refs. 9 and 10 and added more citations as you suggested.

3) p.2 paragraph 2: "block of amplitudes" is rather vague, it should be related to "communication (QAM) symbols" (mentioned in the previous paragraph) better.

Response: We have modified this phrase as you suggested.

4) p.3 paragraph 2: 4 consecutive amplitudes → one DP symbol: This is discussed extensively in [4], I would suggest referencing it.

Response: We have added citations to related previous works.

5) p.3: $\|\cdot\|^2$ is not defined

Response: Its definition has been added to the revised manuscript.

6) p.3: *Explanation of sphere shaping can be connected to the related literature more strongly*

Response: We have improved the explanation of sphere shaping in various places with added citations to previous works. Specifically, we added explanations of sphere shaping in relation to existing works in the second paragraph of Page 2, at the end of Page 4, and in the “Sphere shaping” section in Page 13.

7) p.3 paragraph 1: *“(...) if every shaped block is plotted as a point in 4n-dimensional signal space, with the i-th amplitude being the position of the point on the i-th coordinate axis, the points uniformly fill a 4n-dimensional (hyper-)sphere (...)”* → *Since the amplitudes are drawn from a discrete set, this is not correct. The points are located in a 4n-dimensional spherical region of a 4n-dimensional rectangular lattice. I would avoid the use of the phrase “uniformly fill”.*

Response: Thanks for pointing out the inaccuracies in the description. We have improved the description of sphere shaping as follows: “the points are distributed uniformly over a set of 4n-dimensional square lattice points that lie on or contained in a 4n-dimensional (hyper-) sphere of radius $\sqrt{E_{\text{Shaped}}^*}$ (due to the symmetry by equiprobable signs)”

8) p.4: *I would recommend the use of R instead of H for the information rate, since H is almost always used to denote the entropy*

Response: We have changed the notation from H to R throughout the paper.

9) p.4 Fig. 1(a): *Sphere shaping indeed decreases the maximum energy substantially. However, what is more important is the decrease in average energy. I would recommend indicating the decrease in average energy, and even relating it to the “linear” shaping gain discussed, e.g., in [9].*

Response: To address this concern, we added two sentences to Page 4 with proper citations as “The average energy of \mathbf{a} to achieve R with sphere shaping decreases with increasing block length $4n$ (see, e.g., ^{8,16}), achieving a theoretical minimum average energy as $n \rightarrow \infty$. We refer to the reduction in average energy of \mathbf{a} by shaping as the fundamental shaping efficiency in this article.” The fundamental shaping efficiency is mentioned in Page 7 when explaining the NGMI of Fig. 2(b), and in Page 11 when analyzing Figs. 5(c) and (d). The reduction in average energy by sphere shaping is important (as is well known for communications over linear transmission media), but for nonlinear optical fiber communications that is the focus of this study, we would like to stress that the reduction in maximum energy (more precisely, the reduction in high-order statistical moments) is just as important as the average energy. Compared to infinite-length sphere shaping, the sphere shaping with $n = 5$ can produce 0.9 to 1.0 dB higher SNR_{Eff} after nonlinear fiber propagation (see Fig. 2(a)) while achieving 0.6 dB lower fundamental shaping efficiency.

10) p.4 Fig. 1(b): *I do not understand what this figure tries to explain, or how Fig. 1(a) relates to this.*

Response: Fig. 1(a) shows only the *probability distribution* of total energy in unshaped versus shaped blocks, without the context of communications over a physical medium. Fig. 1(b) shows in the context of communications over a physical medium, *in which rectangular regions* of the time-frequency plane such probability distributions can be measured. The size of the rectangular regions is determined by R_{Sym} and n , and the distribution within this region is determined by n for the given \mathcal{A} and R . We believe that how Fig. 1(a) relates to Fig. 1(b) is expressed in the current statement, which reads “In a densely packed WDM system with identical channel configurations, such probabilistic energy distributions as in Fig. 1(a) are observed within each rectangular block that divides lightwaves in the time-frequency plane as shown in Fig. 1(b), where the width and height of the block are determined by both R_{Sym} and n .” To further clarify why this relation matters for communications, we added “While the shaping block length n or the distribution of energy (cf. Fig. 1(a)) has been optimized in existing studies^{8,11-12,15-17,23} to mitigate nonlinear interference (NLI), the spectro-temporal region where the distribution is found (cf. Fig. 1(b)) has never been noted previously. In the following sections, we will see that it is the distribution of light energy in all aspects of probability, time, and frequency that determines the manifestation of Kerr nonlinearity as NLI, and thus R_{Sym} and n must be controlled simultaneously to minimize NLI.”

11) p.4: sentence before (1) and (2): “(...) η increases as the central moment (...)” $\rightarrow \eta$ or P_{NLI} ?

Response: We believe that the current statement is correct as is. η increases as the *central moment* μ_n of p increases. By (1), $\langle P_{NLI} \rangle$ also increases as μ_n increases.

12) p.5: “(...) deviation by the instantaneous power (...)” \rightarrow “(...) deviation of the instantaneous power (...) from average $\langle p \rangle = 1$ (...)” ?

Response: We corrected this phrase as “deviation using the instantaneous power”.

13) p.5: “(...) appears to be more spread around the average (...)” \rightarrow I think this can be related to the difference in their $\bar{\mu}_2$, right? If so, please refer to these values.

Response: We rephrased this sentence as “sphere shaping appears to make the instantaneous power more spread out around the average power (note the increase of μ_2 from 0.32 to 0.687 after sphere shaping)”.

14) Figures 1(c-d): What is the shaping rate and the resulting distribution?

Response: We added “All shaped symbols in Fig. 1 achieve $R = 6.4$ ” to the caption of Fig. 1. The probability distribution of energy in a block of 5 symbols is shown in Fig. 1(a). The probability distribution of normalized energy in each symbol is shown in Fig. 1(c).

15) p.6: I do not understand what “With our system settings, $n = 5120$ approximately represents i.i.d. shaping from the dispersion perspective, since each symbol is dispersed over less than 5120 symbol periods in all conditions except for $R_{Sym} = 88$ GBd in Link D.” exactly means.

Response: It meant that the channel memory due to chromatic dispersion is shorter than the shaping block length, except for $R_{Sym} = 88$ GBd in Link D. However, in the revised manuscript, we additionally provided results obtained with i.i.d. random symbols drawn from a Maxwell-Boltzmann distribution. Accordingly, the sentence you mentioned has been deleted.

16) p.6: *I am not sure how the observation (iii) is explained with “(...) since increasing R_{Sym} reduces the shaping block duration in absolute time for the same n ”.*

Response: To address your concern (and as Reviewer 2 suggested), this sentence has been modified to “(iii) the optimal R_{Sym} increases as n decreases (compare, e.g., the yellow and red stars).”

17) p.7: *I understand the observation “Areas far from the circled areas have a negligible impact on SNR_{Eff} . But I believe this can be related to Figs. 2(a,c) more strongly and quantitatively.*

Response: We are not sure if you wanted to refer to Figs. 2(c-5) and (c-8). It is apparent from these figures that the low symbol rate region in Fig. 2(c-5) and high symbol rate region in Fig. 2(c-8) have a negligible impact on SNR_{Eff} . Taking η^∞ of these figures for i.i.d. shaping as baselines, $\Delta\eta$ in Figs. 2(c-6) and (c-9) is added as offsets for finite-length shaping, and here we want to say that how significant the effect of such offsets $\Delta\eta$ on SNR_{Eff} is determined by the baseline η^∞ . The meaning of the sentence is apparent from its preceding sentence, which reads “the influence of $\Delta\eta_{SPM}$ and $\Delta\eta_{XPM}$ on SNR_{Eff} is prominent only near the red circled areas, where their base coefficients η_{SPM}^∞ and η_{XPM}^∞ are large.”

18) Fig. 3(a): *How many spans?*

Response: Thanks for pointing out the missing information. We added “at 240 spans in Link D” to the figure description in the text and “obtained at 240 spans in Link D” to the figure caption.

19) p.8: *You say “optimal sphere shaping” more than once in the paper. What exactly do you mean by “optimal”? The combination of n and R_{Sym} that maximizes the performance? There is an optimal window length w that maximizes the correlation b/w the windowed moment and the effective SNR. But I am not sure what you mean by “optimal” sphere shaping. Clarification is needed.*

Response: We made sure that the meaning of the optimality is clear in all places. In the first sentence of the subsection “Demonstration of Optimal Sphere Shaping through Full C-Band Transmission Experiment” in Page 9, we clarified that optimal sphere shaping implies maximizing NDR. We also clarified this in Fig. 5 caption. In all other places, we believe that the meaning of optimality for sphere shaping or system parameters is explicitly stated or apparent from its surrounding sentences.

20) p.9: *I believe the conclusion “In general, n^* increases as $|D_{Total}|$ (green solid line) increases, (...)” is a bit strong to be made from Fig. 5(a). The optimal block length here is mostly constant around 10-20,*

increasing around 195 THz, but then decreasing again. If the authors can provide some additional explanation for this behaviour, it would make the paper stronger.

Response: We relaxed the conclusion by saying “In general, n^* tends to increase as $|D_{Total}|$ (green solid line) increases.” We attribute the unsmooth curves and deviation from theoretic prediction in Fig. 5 mostly to the difficulty of creating an ideal experimental setup with limited equipment. In particular, the recirculating loop that is the only experimental method of long-haul transmission involves many variations of real-world components. The most important challenge for us was minimizing the optical power excursion across the C-band using the DGEs that have only 0.1 dB nominal attenuation granularity. In true straight-line long-haul systems, the gain tilts and ripples of inline EDFAs can nearly perfectly be flattened by passive optical attenuators with continuous attenuation profiles that were predetermined for a fixed total optical power. On the other hand, in the recirculating loops operated with various total optical powers, the gain tilts and ripples should be flattened by the DGEs with coarse attenuation granularity. A small residual power excursion due to this limitation increases as it accumulates over the number of loops, leading to deviations from the ideal flat optical power spectral density. To address this, we added “The power excursions due to experimental constraints (e.g., a small power excursion caused by coarse attenuation granularity of the DGEs results in an increasing power excursion as the number of loops increases) are considered to be the most important contributor to discrepancy in validation of theory” at the end of Page 10.

21) p.9: At which block length Fig. 5(b) is plotted? Also, similar to my previous comment, the conclusion “(...) R_{Sym} decreases (...) when $|D_{Total}|$ increases (...)” is a bit strong to be made from Fig. 5(b). Instead of simply saying Figs. 5(a,b) agree with Figs. 2(b,a), possible explanations for the peculiar behaviours in Figs. 5(a,b) should be provided.

Response: For clarification, we combined the descriptions of Figs. 5(a) and (b) in the text as “Figs. 5(a) and (b) show, respectively, the optimal n^* and R_{Sym}^* that jointly maximize NDR in each channel,” and added “for maximum NDR” to the captions of Figs. 5(a) and (b). We also relaxed the conclusion by saying “ R_{Sym}^* tends to decrease.” The non-smooth change of optimal parameters is attributed to experimental limitations, and the added sentence for Fig. 5(a) mentioned above also provides an explanation for Fig. 5(b).

22) p.9, third line: SNR_{Eff} of Fig. 4(c), or of Fig. 5(c)?

Response: We corrected this typo.

23) p.11: CCDM is not a sphere shaping algorithm.

Response: We added “Note that CCDM can only approximately realize sphere shaping for finite block lengths, but it converges to ideal sphere shaping with a decreasing approximation error as the block length increases” to the first paragraph in the Methods section.

24) Similar gains in data rate/SNR/AIR to the ones that are reported at the end manuscript exist in the literature. A short review of such works would make the paper stronger.

Response: At the end of the “Demonstration of Optimal Sphere Shaping through Full C-Band Transmission Experiment” section, we added brief explanation about the previous works and emphasized the difference of this work compared to the previous works as “While the previous works^{8,11,12,16,23} optimized only n to observe some gains in SNR_{Eff} and NDR over specific links (e.g., for single-span links), joint optimization of n and R_{Sym} in this work produces significantly larger gains and allows these gains to be achieved over a much wider variety of links.” Due to the word count limit of Nature Communications, this level of explanation seems to be the best we can do.

REFERENCES

- [1] K. Wu, G. Liga, A. Sheikh, F. M. J. Willems, and A. Alvarado, “Temporal energy analysis of symbol sequences for fiber nonlinear interference modelling via energy dispersion index,” Feb. 2021. [Online]. Available: <https://arxiv.org/abs/2102.124114>
- [2] P. Skvortcov, I. Phillips, W. Forsyiaik, T. Koike-Akino, K. Kojima, K. Parsons, and D. S. Millar, “Nonlinearity tolerant LUT-based probabilistic shaping for extended-reach single-span links,” *IEEE Photon. Technol. Lett.*, vol. 32, no. 16, pp. 967–970, 2020.
- [3] T. Fehenberger, “On the impact of finite-length probabilistic shaping on fiber nonlinear interference,” in *Proc. Signal Process. in Photon. Commun. (SPPCom)*, Washington, D.C., United States, July 2020.
- [4] P. Skvortcov, I. Phillips, W. Forsyiaik, T. Koike-Akino, K. Kojima, K. Parsons, and D. S. Millar, “Huffman-coded sphere shaping for extended-reach single-span links,” *IEEE J. Sel. Topics Quantum Electron.*, vol. 27, no. 3: 3500215, May-June 2021.
- [5] S. Civelli, E. Forestieri, and M. Secondini, “Interplay of probabilistic shaping and carrier phase recovery for nonlinearity mitigation,” in *Proc. Eur. Conf. Opt. Commun. (ECOC)*, Brussels, Belgium, Dec. 2020.
- [6] T. Fehenberger, A. Alvarado, G. Böcherer, and N. Hanik, “On probabilistic shaping of quadrature amplitude modulation for the nonlinear fiber channel,” *J. Lightw. Technol.*, vol. 34, no. 21, pp. 5063–5073, Nov. 2016.
- [7] Y. C. Gültekin, A. Alvarado, O. Vassilieva, I. Kim, P. Palacharla, C. M. Okonkwo, and F. M. Willems, “Kurtosis-limited sphere shaping for nonlinear interference noise reduction in optical channels,” June 2021. [Online]. Available: <https://arxiv.org/abs/2105.14794>
- [8] P. Schulte and F. Steiner, “Divergence-optimal fixed-to-fixed length distribution matching with shell mapping,” *IEEE Wireless Commun. Lett.*, vol. 8, no. 2, pp. 620–623, Apr. 2019.
- [9] Y. C. Gültekin, W. J. van Houtum, A. Koppelaar, and F. M. J. Willems, “Enumerative sphere shaping for wireless communications with short packets,” *IEEE Trans. Wireless Commun.*, vol. 19, no. 2, pp. 1098–1112, Feb. 2020.

Response to Reviewer 2:

This work studies the optimization of the time and bandwidth over which sphere shaping should be performed to minimize the impact of nonlinear effects and maximize performance in optical fiber communication.

The work is timely, well written, and reports some novel interesting results.

The optimization of the shaping block length in the nonlinear regime has been already investigated in some other papers but, to my knowledge, the simultaneous optimization of block length and symbol rate (i.e., time and bandwidth) is studied here for the first time. Moreover, this study encompasses several scenarios and a wide range of parameters, both numerically and experimentally.

The methodology is sound and the methods are described with enough detail. Moreover, the conclusions are supported both by numerical and experimental results.

In general, I believe that the paper deserves publication. However, there are some weaknesses and some issues that should be considered and possibly addressed, as detailed below.

Response: Thank you very much for the positive comments and suggestions for improving the paper. We tried to address all of your concerns in the revised manuscript, and hope that the weaknesses and issues found in the previous manuscript have been removed by this revision.

1) As a general note, while reading the paper, I was expecting to find a theoretical analysis which explains and predicts the results observed numerically and experimentally. This expectation is not fully met by the paper, which provides only some insight about the observed phenomena and some empirical laws to model them - e.g., (7), (8) and the use of the windowed moments in the EGN model. In fact, there are a number of perturbation models in the literature which could be used to attempt such an analysis, for instance by removing the i.i.d. assumption in the EGN model. An example can be found in Liga et al "Extending Fibre Nonlinear Interference Power Modelling to Account for General Dual-Polarisation 4D Modulation Formats", Entropy, 2020. I recommend discussing this issue and highlighting the empirical nature of the proposed model.

Response: Thank you for referring us to the previous work on propagation modeling of structured lightwaves. In fact, after submitting the first manuscript, we learned from an author of Ref. 26 (which is the paper that you mentioned above) at a conference that an extended EGN model was published last year for the presence of structures (or correlations) in four amplitudes, i.e., in one symbol period. To the best of our knowledge, this (and its simplification in Ref. 27) is the only published work on propagation modeling of structured lightwaves. However, given the enormous complexity of the mathematical expansion to account for only four correlated amplitudes in Ref. 26, it can be inferred that it is daunting to find a mathematically accurate analytical model to handle tens to thousands of correlated amplitudes that we cover in this paper. In this regard, we addressed your concern by adding a sentence to the beginning of Page 3 as “Analytical approaches to take into account the structure of lightwaves have so far been successful up to one symbol^{26,27}, but extending the analysis to structures spanning many symbols seems mathematically daunting. To quantify the effect of large temporal structures of lightwave on Kerr nonlinearity, empirical approaches are being taken in rapidly growing recent studies^{8,15-17,23}. However, there has been no study on *whether* or *how* the symbol rate affects this quantification.” We also stated on Page 9 that “The EGN model assumes i.i.d. amplitudes and phases of symbols, and hence is not accurate

for lightwaves with local energy structures. There is a recently developed analytical model^{26,27} that extends the EGN model to account for energy structures present over one symbol period, but extending this further to energy structures spanning tens to thousands of symbol periods that we deal with in this work seems mathematically intractable. Therefore, we allow for model mismatch by using the classical EGN model, but improve the accuracy of evaluating structured lightwaves (green solid lines in the figure) by replacing μ_2 and μ_3 with optimized $\bar{\mu}_2$ and $\bar{\mu}_3$ (i.e., obtained with w_{SPM}^* and w_{XPM}^*).

2) As a second general comment, the analysis practically neglects the impact of carrier recovery. Indeed, this might be substantial, as a sufficiently fast carrier recovery algorithm can mitigate the impact of nonlinear phase noise and change the dependence of performance on block length (see for instance Civelli et al. "Interplay of probabilistic shaping and carrier phase recovery for nonlinearity mitigation", ECOC 2020). With carrier recovery, there might be no advantage in using short (optimized) block lengths. It seems that carrier recovery is not present in the simulations and is quite slow in the experiments (using only 1 pilot every 48 symbols), possibly overestimating the advantage of using a short optimized block length. Please comment about that. As a side note, for the sake of transparency, I am a coauthor of the above mentioned paper. I do not want to push my own work, therefore feel free to decide if the paper is relevant or not to your work and if it deserves to be cited or not. This decision will not affect my final recommendation.

Response: Thanks for pointing out another prior work related to this manuscript and for providing transparency. As reported in the suggested reference paper, the phase rotation due to NLI can be mitigated by the BPS algorithm under certain conditions. To check this, we performed the BPS on our simulation data, but no appreciable effect was observed in our settings, especially over long distances with low SNR as shown in the table below for Link D. The number of symbols, $2N_{BPS} + 1$, that are used for averaging the phase rotation is determined such that the resulting SNR_{Eff} is approximately maximized, and a launch power of 1 dBm is used that is close to optimal.

Number of Spans	$2N_{BPS} + 1$	$N_{SC} = 1$		$N_{SC} = 32$	
		Without BPS	With BPS	Without BPS	With BPS
15	17	20.41	20.57	20.88	20.94
240	513	7.34	7.38	8.39	8.31

Similarly, when we performed the BPS on our experimental data, we did not observe noticeable change in SNR_{Eff} . Note that for carrier recovery of the experimental data, we used linear interpolation to obtain the phase of 47 payload symbols between 2 consecutive pilot symbols. We briefly discussed about the BPS at the end of the Discussion section as "Also, the use of advanced carrier recovery algorithms such as the maximum-likelihood blind phase search (BPS) may influence the impact of sphere shaping on NLI under certain conditions⁴⁶, but in this work at transmission distances that match the sphere-shaped 16-QAM format, no noticeable effect was observed using the BPS."

3) Abstract: The sentence "In optical fiber, however, the sphere shaping induces Kerr nonlinearity in a peculiar way that makes analysis of transmission performance difficult, potentially lowering the communications capacity" sounds a bit odd. It suggests that it is sphere shaping that makes the analysis

difficult, with Kerr nonlinearity just playing an indirect role. In fact, I think the opposite is true. Moreover, it suggests that the difficulty of the analysis may play a role in the capacity reduction.

Response: To emphasize that it is Kerr nonlinearity that is hard to understand, we rephrased the sentence as “**However, when shaped lightwaves are transmitted through optical fiber, Kerr nonlinearity manifests itself as nonlinear interference in a peculiar way, potentially lowering communications capacity.**”

4) Introduction: The idea of using sphere shaping on an optimized block length to minimize the impact of nonlinear interference was first proposed in [10] and then further investigated by some of the same authors in Geller et al. "A Shaping Algorithm for Mitigating Inter-Channel Nonlinear Phase-Noise in Nonlinear Fiber Systems", JLT 2016. Your work (as well as some other recent works) heavily relies on this idea, extending the optimization to the frequency domain. I think that it would be fair to better acknowledge this fact and the role of these two papers.

Response: We clarified what has been done in the previous works and what is the difference of this work compared to the previous works by adding several sentences to the paper as follows. In the second paragraph of the Introduction section, we added “**For this reason, there have been several recent approaches to optimizing the shaping block length to mitigate Kerr nonlinearity^{8,11-12,15-16,23}.**” At the end of the “Sphere Shaping of Lightwaves” section, we added “**While the shaping block length n or the distribution of energy (cf. Fig. 1(a)) has been optimized in existing studies^{8,11-12,15-17,23} to mitigate nonlinear interference (NLI), the spectro-temporal region where the distribution is found (cf. Fig. 1(b)) has never been noted previously.**” At the end of the “Demonstration of Optimal Sphere Shaping through Full C-Band Transmission Experiment” section, we added “**While the previous works^{8,11,12,16,23} optimized only n to observe some gains in SNR_{Eff} and NDR over specific links (e.g., for single-span links), joint optimization of n and R_{Sym} in this work produces significantly larger gains and allows these gains to be achieved over a much wider variety of links.**”

As for the last sentence added above, the findings of this article suggest that for symbol rates greater than 10 GBd, as is commonly used in recent experiments, optimization of the shaping block length is only effective over very short distances on dispersion-unmanaged SSMF links where net dispersion is small (see equations (7), (8) and Supplementary Fig. 4). This perhaps explains why most of the previous works on optimizing the shaping block length used single-span links. For long distances on dispersion-unmanaged SSMF links, we are able to observe a noticeable benefit by shaping block length optimization only when using a low symbol rate of a few GBd. The last sentence above emphasizes this briefly.

5) Page 3, Results: you state that "The symmetry of the probability allows for legitimate analysis with only positive amplitudes, and hence we omit the sign throughout this article". I guess that the presence of a random sign with uniform distribution is implicitly assumed in the paper and neglected only in the description, whereas it is included both in the simulations and in the experiments. Is that correct? I am not sure that transmitting only positive amplitudes would indeed give the same results. I think that two consecutive symbols with the same sign or with opposite signs induce a fundamentally different response in the nonlinear channel. Please be more explicit.

Response: You're correct that random signs were used with equal probability in simulations and experiments. We have clarified this by rewriting the sentence as "we omit the sign throughout this article for descriptive purposes (but in simulations and experiments, equally distributed positive and negative signs are used)."

6) Page 3, Results: the sentence "one dual-polarization symbol that maintains the value over a period of T_{sym} (s) and can change the value at a rate of $R_{sym}=1/T_{sym}$ (Bd)", suggests a rectangular pulse shape. I suggest revising the sentence. Moreover, the indication of the units of measure in this way is neither standard nor required.

Response: To eliminate the confusion that the pulse shape is rectangular, we have rephrased the sentence as "In our system, four consecutive amplitudes constitute one dual-polarization symbol, as has been done, e.g., in ^{16,17}, that is transmitted with a symbol period of T_{Sym} and a symbol rate of $R_{Sym} = 1/T_{Sym}$." We also removed the unnecessary unit of T_{Sym} and R_{Sym} here.

7) Page 5, 4 lines before the end of the subsection: The sentence "As w increases, μ_2 remains the same for all types of i.i.d. symbols" is misleading. It is not "the same" for all types of symbols but, rather, it remains "constant" as w increases for a given modulation (type of symbols).

Response: We fixed the incorrect sentence as " μ_2 remains constant for i.i.d. symbols."

8) Equation (3): for sphere shaping, the transmitted signal is a cyclostationary process with period equal to the length of the shaped blocks (unless a uniform random delay is assumed). Therefore, I guess that the windowed moments depend on the time at which the window is centered, unless they are averaged over it. Please comment about that and explain if and how you have considered this time dependence in your analysis.

Response: You're correct that sphere-shaped signal, and hence ρ , is a cyclostationary process with a period equal to the n symbol periods. If the sliding step size of the moving average filter $\langle \cdot \rangle_w$ is greater than one symbol, it is also true that the windowed moments $\bar{\mu}_n$ in equation (3) can have different values depending on where the window starts sliding in the stream of the shaped symbol blocks. However, we use a sliding step size of one symbol, as is typically done in moving average filters. In this case, $\bar{\mu}_n$ can have only a single fixed value. To avoid potential confusion, we explicitly stated this fact under equation (3) as " $\langle \cdot \rangle_w$ denotes a moving average filter with a sliding window of length w symbols (with a sliding step size of one symbol)."

9) Page 5, Section "Optimization of Sphere Shaping in the Time-Frequency Plane": Please specify what QAM format is considered in this section. I guess from the given data that it is 16QAM (as in the previous section). Yet, an explicit mention here would be useful.

Response: In the "Optimization of Sphere Shaping Parameters in the Time-Frequency Plane" section, we explicitly stated the modulation format by modifying a sentence to "Sphere shaping is performed in each

channel with $n = 5, 10, 20, 40, 80, 320, 1280, 5120$, with a fixed $R = 6.4$ bits per dual-polarization symbol using 16-QAM.” Also, in the “Demonstration of Optimal Sphere Shaping through Full C-Band Transmission Experiment” section, we specified the modulation format by modifying a sentence to “For each pair of n and R_{Sym} , the NDR achieved by sphere shaping of 16-QAM is determined by...”.

10) Page 6: You state that "With our system settings, $n=5120$ approximately represents i.i.d. shaping from the dispersion perspective, since each symbol is dispersed over less than 5120 symbol periods in all conditions except for $R_{sym}=88$ Gbd in Link D" and that "like the i.i.d. uniform QAM, the optimal R_{sym} for i.i.d. shaping to maximize SNR_{eff} (yellow stars) decreases as the total net dispersion D_{Total} increases". I think that this discussion is a bit weak and that the analogy with the symbol rate optimization for uniform i.i.d. QAM is misleading for two reasons: 1) The dependence of SNR_{eff} on symbol rate for the uniform i.i.d. QAM case vanishes for Gaussian symbols (the EGN and GN model converge to the same equations, see for instance [23]). Therefore, I expect a very little dependence on the symbol rate for shaped i.i.d. symbols, as they approximate i.i.d. Gaussian symbols. In fact, the dependence on symbol rate in Fig. 2a for the case $n=5120$ is very weak (and would be probably even weaker for shaped 64-QAM or 256-QAM) and the peak SNR is not so different from the surrounding values in the simulation range; 2) assuming that $n=5120$ represents the i.i.d. case from a dispersion perspective is an approximation whose accuracy decreases as the symbol rate increases; therefore, it cannot be used when explaining the (weak) dependence of SNR_{eff} on symbol rate. In fact, for $n=5120$, this dependence becomes a little more relevant exactly where the approximation becomes looser. In my opinion, it seems more likely that the dependence of SNR_{eff} on symbol rate is mainly explained, even for $n=5120$, by the same reason that explains it for lower n .

Response: Thank you very much for this valuable insight. To address your concern, we added split-step simulation results of i.i.d. shaping, in which shaped symbols are drawn from a 16-QAM alphabet according to a Maxwell-Boltzmann distribution. Inclusion of i.i.d. shaping posed difficulties in visualizing SNR_{Eff} in contour plots of Fig. 2 and Supplementary Figs. 1-4, since it is equivalent to infinite-length sphere shaping in principle. In fact, as mentioned in the Methods section, the number of transmitted symbols in each WDM channel is limited to $2^{18}, 2^{17}, 2^{16}, 2^{15}, 2^{14}, 2^{14}, 2^{14}$ dual-polarization symbols in each channel, respectively, for $N_{Ch} = 1, 2, 4, 8, 16, 32, 64$. Nevertheless, we kept the format of the contour plots the same as in the first manuscript, adding one point at the top of each R_{Sym} . This visualization would be acceptable as we see that the difference in SNR_{Eff} between $n = 5120$ and i.i.d. shaping is very small in all figures. Accordingly, below equation (4), we explained the figure as “The top points at each R_{Sym} represent i.i.d. shaping, so their y-axis values are not exact values but merely represent very large numbers. The y-axis values for all other points are exact”.

We agree that SNR_{Eff} is not changed by R_{Sym} if the modulation format is continuous Gaussian, and that the dependence of SNR_{Eff} on R_{Sym} will be very weak in the case of sphere shaping with high-order QAM. However, with relatively small i.i.d. shaped 16-QAM, we observe about 1 dB change in η^∞ due to R_{Sym} , which is not much different from the change in η due to R_{Sym} in i.i.d. uniform QAM (see the lower figure in Fig. 4 of Ref. 25). To emphasize the small QAM order, we specified the modulation format as “(i) like the i.i.d. uniform 16-QAM²⁵, the optimal R_{Sym} for i.i.d. shaping of 16-QAM to maximize SNR_{Eff} (yellow stars) decreases as the total net dispersion D_{Total} increases”. Also, we wrote “The

influence of R_{Sym} on η^∞ is expected to decrease further as the modulation order increases and the shaped signal approaches continuous Gaussian” in the first paragraph of Page 8, and “It also remains for future work to see how the dependence of η on R_{Sym} and n changes as the sphere-shaped QAM modulation order increases to approach continuous Gaussian signaling in terms of the time-averaged probability distribution.” in the first paragraph of the Discussion section.

11) Page 6, you state that "the optimal R_{sym} is higher for finite-length shaping than for i.i.d. shaping (compare the yellow and red stars), since increasing R_{sym} reduces the shaping block duration in absolute time for the same n ". Following up on the previous comment, can we better (and more simply) say that the optimal symbol rate increases when n decreases?

Response: Thanks for the great suggestion. Following your suggestion, we have modified the sentence as “(iii) the optimal R_{Sym} increases as n decreases (compare, e.g., the yellow and red stars).”

12) Still on the same matter: you refer many times in the results to the i.i.d. case, considering $n=5120$ as an approximation to it. Why not directly simulating the true i.i.d. case? It should be sufficient to draw i.i.d. QAM symbols from a Maxwell-Boltzmann distribution. That would be a more meaningful benchmark against which to compare the results for finite block lengths and, for instance, to compute η^∞ in (6).

Response: We agree that the inclusion of i.i.d. shaping would greatly improve the paper, and as mentioned above, we added split-step simulation results of i.i.d. shaping. Figures and many parts of the text have been modified accordingly.

13) Page 8, Demonstration of Optimal Sphere Shaping through Full C-Band Transmission Experiment: The experiment is performed on a link that is substantially different from the 4 links considered in the simulation. Why? In particular, each channel sees a different dispersion profile, making the interpretation of the results harder. Moreover, the important dispersion-unmanaged case is not considered in the experiments. Is there a particular reason for this choice? Is it instrumental to highlight some effect or behavior? Please explain and motivate this choice.

Response: Although we aim to estimate the benefits of shaping optimization in the full C-band transmission, our simulation setup is limited to a total bandwidth of 100 GHz due to the long simulation time. The experiment is much easier to implement the full C-band transmission setup, but it is very difficult to test all 4 links through the experiment due to the availability of different types of fiber and the enormous effort required to construct a link (e.g., custom gain flattening for each link and for each launch power is a huge effort). On one hand, performing the experiment in a single link close to one of the 4 links used in simulation is good for validating the findings of simulation. But it has some weaknesses as follows. (i) The validation is limited to a specific dispersion coefficient of the selected link. (ii) To evaluate the impact of dispersion on NLI, the transmission distance needs to be changed as shown in, e.g., Supplementary Fig. 4. However, as the distance changes, 16-QAM becomes no longer a suitable modulation format that matches the underlying SNR_{Eff} , making it difficult to evaluate the NDR increase

achieved by optimizing the sphere shaping parameters with the same modulation format as in the simulation.

On the other hand, allowing for mismatches of the setup between simulation and experiment, performing the experiment on a dispersion-managed link near zero-dispersion frequency with a dispersion slope allows us to see how NLI varies with the sphere shaping parameters under various chromatic dispersion coefficients, even at a fixed distance. This allows for qualitative validation of simulation results, as briefly mentioned in the first paragraph of the “Demonstration of Optimal Sphere Shaping through Full C-Band Transmission Experiment” section as “The dependence of the optimal sphere shaping on dispersion is conveniently verified in a recirculation loop that accumulates varying dispersions over frequency.” While we agree that the dispersion-unmanaged system is an important use case, we would like to leave the experiment in dispersion-unmanaged systems for future work.

14) Page 11, Methods: you state that "The digital sphere shaping encoder is implemented by enumerative sphere shaping (ESS) for $n = 5, 10, 20, 40, 80$, and by constant composition distribution matching (CCDM) for $N = 320, 1280, 5120$." and "...consume 0.583, 0.346, 0.209, 0.137, 0.099, 0.025, 0.009, 0.002 dB more average symbol energy than ideal shaping, respectively". Strictly speaking, CCDM does not implement sphere shaping (as all constant composition sequences lie on the surface of the sphere, partially covering it) and, in fact, its energy loss is higher than that of ESS for the same blocklength. Of course, they both asymptotically converge to i.i.d. Maxwell-Boltzmann symbols, but as you are studying how performance changes with block length, you cannot assume that such an asymptotic regime has been achieved. In fact, I am also a bit surprised that CCDM with $n=320$ performs significantly better than ESS with $n=80$ (see, e.g., the comparison in [7]). Please comment about that and double check the numbers (I have not verified them, it is just a feeling and I might be wrong).

Marco Secondini

Response: As you pointed out, CCDM can only approximately realize sphere shaping. Its rate loss decreases as the block length increases, and approaches to ideal sphere shaping as the block length tends to infinity, as a result of the sphere hardening phenomenon. To address your concern, in the first paragraph of the Methods section, we added the sentence “**Note that CCDM can only approximately realize sphere shaping for finite block lengths, but it converges to ideal sphere shaping with a decreasing approximation error as the block length increases.**”

To explain the performance of CCDM as good as that of ESS at $n = 320$, we have reproduced Fig. 3 of Ref. 8 (which is [7] in your reference number) in the figure on the left below with $R = 1.6$ bit/amp and 64-QAM. However, unlike Ref. 8, this paper uses 16-QAM, and in this case the increase in rate loss of CCDM compared to ESS is not large even at a fairly short block length, as shown on the right figure below. This explains the good performance of CCDM at $n = 320$.

REVIEWERS' COMMENTS

Reviewer #1 (Remarks to the Author):

All my comments and suggestions seem to be taken into account. I can suggest the publication of the paper following a proofreading.

Reviewer #2 (Remarks to the Author):

All my comments have been properly addressed.

Marco Secondini

**Response to Reviewers' Comments on NCOMM-21-15776
"Shaping Lightwaves in Time and Frequency for Optical Fiber
Communication"**

Junho Cho, Xi Chen, Greg Raybon, Di Che, Ellsworth Burrows, Samuel Olsson, and Robert Tkach

Dear Reviewers,

We sincerely thank all reviewers for carefully reading and commenting on our paper again. Please find our point-by-point response to the comments below, where the reviewers' comments are reproduced verbatim in *blue italic type*, and our response is written in black roman type.

Response to Reviewer 1:

All my comments and suggestions seem to be taken into account. I can suggest the publication of the paper following a proofreading.

Response: Your comments have greatly improved our manuscript. Thank you again very much for this.

Response to Reviewer 2:

All my comments have been properly addressed.

Response: Thank you very much for your comments during the last two reviews. They advanced our understanding of the subject area and significantly improved the manuscript.